# A conserved cell division protein directly regulates FtsZ dynamics in filamentous and unicellular actinobacteria

**Félix Ramos-León[1†], Matthew J Bush[1†], Joseph W Sallmen[1], Govind Chandra[1], Jake Richardson[2], Kim C Findlay[2], Joseph R McCormick[3], Susan Schlimpert[1]\***

[1]Department of Molecular Microbiology, John Innes Centre, Norwich, United Kingdom; [2]Department of Cell and Developmental Biology, John Innes Centre, Norwich, United Kingdom; [3]Department of Biological Sciences, Duquesne University, Pittsburgh, United States

**Abstract** Bacterial cell division is driven by the polymerization of the GTPase FtsZ into a contractile structure, the so-called Z-ring. This essential process involves proteins that modulate FtsZ dynamics and hence the overall Z-ring architecture. Actinobacteria like *Streptomyces* and *Mycobacterium* lack known key FtsZ-regulators. Here we report the identification of SepH, a conserved actinobacterial protein that directly regulates FtsZ dynamics. We show that SepH is crucially involved in cell division in *Streptomyces venezuelae* and that it binds FtsZ via a conserved helix-turn-helix motif, stimulating the assembly of FtsZ protofilaments. Comparative *in vitro* studies using the SepH homolog from *Mycobacterium smegmatis* further reveal that SepH can also bundle FtsZ protofilaments, indicating an additional Z-ring stabilizing function *in vivo*. We propose that SepH plays a crucial role at the onset of cytokinesis in actinobacteria by promoting the assembly of FtsZ filaments into division-competent Z-rings that can go on to mediate septum synthesis.

**\*For correspondence:**
susan.schlimpert@jic.ac.uk

[†]These authors contributed equally to this work

**Competing interests:** The authors declare that no competing interests exist.

## Introduction

Cell division is an essential process for almost all living organisms. The core component of the bacterial cell division machinery is the bacterial tubulin homolog FtsZ, which forms the so-called Z-ring at the future division site. Like tubulin, FtsZ monomers polymerize in a GTP-dependent manner into cytoplasmic filaments that undergo treadmilling *in vivo*, a process in which FtsZ subunits are selectively added to one end and removed from the other end (*Loose and Mitchison, 2014*; *Yang et al., 2017*). The Z-ring provides a dynamic scaffold for the assembly of a multiprotein division machinery, the divisome. In addition, FtsZ treadmilling also guides the circumferential movement of peptidoglycan synthases at the division site which leads to septum formation and cell membrane constriction (*Bisson-Filho et al., 2017*; *Perez et al., 2019*; *Yang et al., 2017*).

While the rate of treadmilling is set by the FtsZ GTPase activity, the overall architecture of the Z-ring is modulated by FtsZ-binding proteins that can influence its positioning, membrane interaction or stimulate FtsZ filament formation and bundling (*Caldas et al., 2019*; *García-Soriano et al., 2020*; *McQuillen and Xiao, 2020*). The function of many of these proteins has been well characterized in the classical rod-shaped model organisms *Escherichia coli* and *Bacillus subtilis*. However, members of the actinobacteria, which include industrially and medically important species such as the prolific antibiotic producers of the genus *Streptomyces*, or the human pathogens *Mycobacterium tuberculosis* and *Corynebacterium diphtheriae*, lack most of the key components that are known to regulate the dynamics of Z-ring formation. This raises the fundamental question as to how exactly the assembly and the architecture of the Z-ring is controlled in these bacteria.

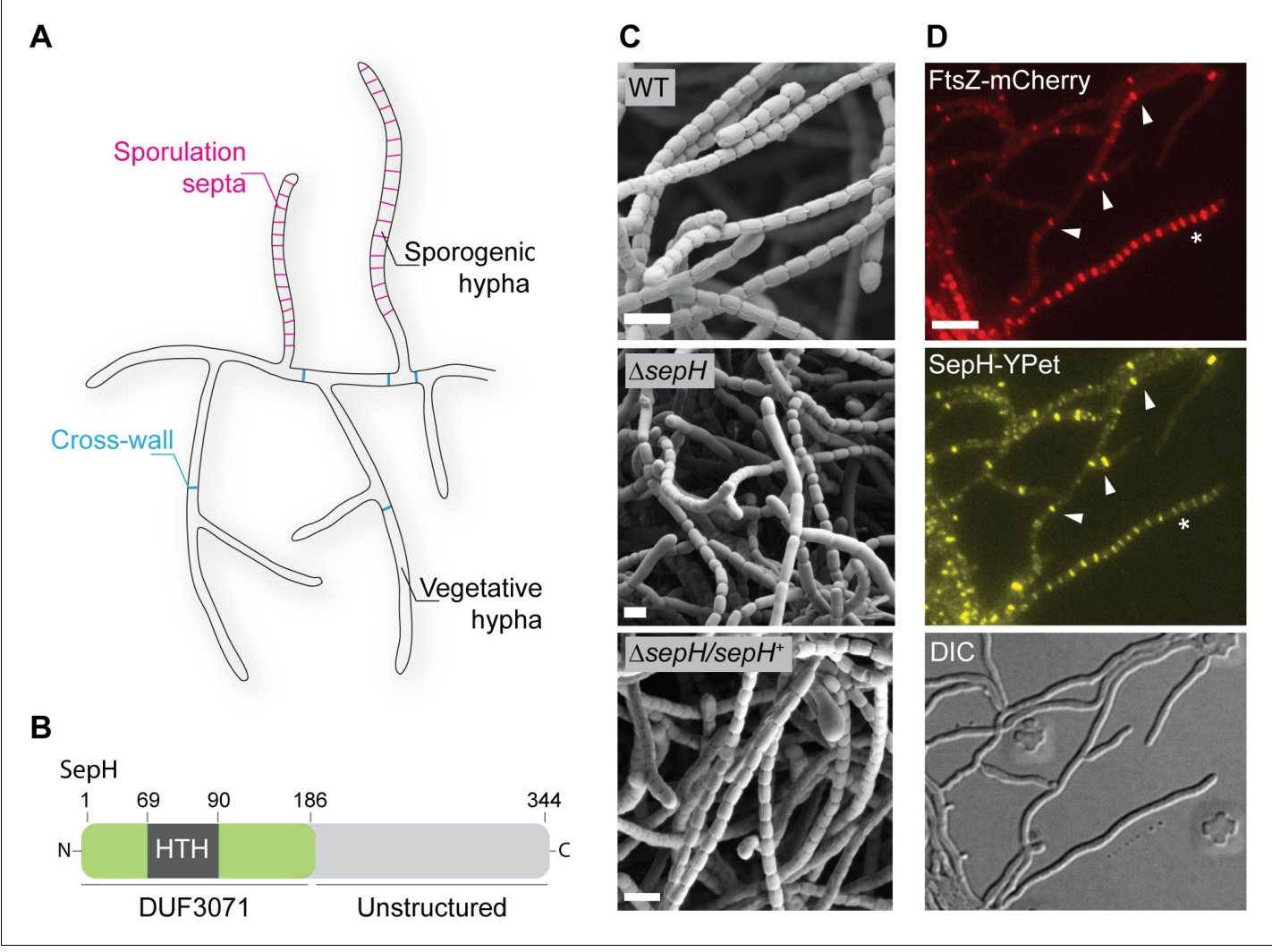

**Figure 1.** *sepH* is required for sporulation septation in *Streptomyces venezuelae*. (**A**) Schematic illustrating the multicellular life style of *Streptomyces* including the two FtsZ-dependent modes of cell division that occur in vegetative and sporogenic hyphae: cross-wall formation and sporulation septation. (**B**) Schematic of the predicted SepH domain organization including the N-terminal DUF3071 domain containing a helix-turn-helix (HTH) motif and the unstructured C-terminal domain. Numbers indicate corresponding amino acid positions. (**C**) Cryo-scanning electron micrographs of sporogenic hyphae from wild-type (WT) *S. venezuelae*, the Δ*sepH* mutant (SV56), and the complemented mutant strain Δ*sepH/sepH⁺* (MB747). Scale bars: 2 μm. (**D**) Subcellular co-localization of fluorescently labeled FtsZ (FtsZ-mCherry) and SepH (SepH-YPet) in vegetative and sporulating hyphae. Fluorescent gene fusions were expressed in the WT background (MB751). White arrow heads point at co-localization at cross-walls in vegetative hyphae and the asterisk denotes a sporogenic hypha undergoing sporulation septation. Scale bar: 5 μm.

The online version of this article includes the following source data and figure supplement(s) for figure 1:

**Figure supplement 1.** *sepH* is a direct target of the transcriptional regulators WhiA and WhiB.

**Figure supplement 2.** Spore length analysis of wild-type (WT) *S. venezuelae*, the Δ*sepH* mutant (SV56), and the complemented mutant strain Δ*sepH/ sepH⁺* (MB747).

**Figure supplement 2—source data 1.** Spore size measurement data.

**Figure supplement 3.** Localization and corresponding protein abundance of SepH-YPet in the wild-type (WT) and the Δ*ftsZ* mutant.

*Streptomyces* are Gram-positive soil bacteria that have a fascinating multicellular life cycle involving filamentous growth and sporulation (*Bush et al., 2015*). Unlike most unicellular organisms that assemble one Z-ring and divide by binary division, *Streptomyces* have two functionally distinct modes of cell division: vegetative cross-wall formation and sporulation septation (*Figure 1A*). Cross-walls divide the growing mycelium occasionally into long multigenomic compartments that remain physically connected. In contrast, during reproductive growth, dozens of sporulation septa are

simultaneously deposited in a ladder-like pattern between the segregating chromosomes along the length of sporogenic hyphae. Sporulation septa eventually constrict, leading to cell–cell separation and the release of unigenomic spores. Both these forms of cell division require FtsZ, but unlike in most other bacteria the *ftsZ* gene can be deleted in *Streptomyces*, leading to viable hyphae that lack both cross-walls and sporulation septa (*McCormick et al., 1994*; *Santos-Beneit et al., 2017*). The *Streptomyces* divisome is comprised of several conserved core divisome components including FtsQ, DivIC, FtsL and the cell wall synthesis proteins FtsI and FtsW (*McCormick, 2009*; *Cantlay et al., 2021*). In addition, the *Streptomyces* cell division machinery includes the membrane anchor SepF, two additional SepF-like proteins of unknown function (SepF2 and SepF3), the two dynamin-like proteins DynA and DynB, which ensure the stability of Z-rings during sporulation, and the actinomycete-specific protein SsgB, which has been proposed to recruit FtsZ to future sporulation septation sites (*Schlimpert et al., 2017*; *Willemse et al., 2011*). However, factors that affect the dynamics of Z-ring formation and regulate its architecture have not yet been identified in actinobacteria and the mechanisms that control cell division in *Streptomyces* and related actinobacteria are poorly understood.

Here, we report the identification and characterization of SepH, a conserved actinobacterial-specific cell division protein that directly binds FtsZ and regulates the dynamics of Z-ring formation in filamentous *Streptomyces* and in rod-shaped *Mycobacterium* species. We find that SepH co-localizes with FtsZ in *Streptomyces* and is required for regular cross-wall formation and sporulation septation. Biochemical characterization of SepH from *Streptomyces venezuelae* and *Mycobacterium smegmatis* revealed that SepH interacts with FtsZ via a highly conserved helix-turn-helix motif and stimulates the formation of FtsZ protofilaments *in vitro*. In addition, SepH from *M. smegmatis* promotes the lateral interaction of FtsZ filaments. Our data suggest that SepH fulfills a crucial function during the initial stages of cell division by increasing the local concentration of FtsZ, thereby stimulating the assembly of division-competent Z-rings.

## Results

### SepH is required for regular sporulation in *Streptomyces venezuelae*

In *Streptomyces*, the initiation of sporulation-specific cell division is controlled by two key transcriptional regulators, WhiA and WhiB. Recent work by Bush et al. determined the regulon of WhiA and WhiB, which co-control the expression of ~240 transcriptional units (*Bush et al., 2016*; *Bush et al., 2013*). To identify novel regulators of Z-ring formation in actinomycetes, we chose to focus on uncharacterized gene products which are conserved across streptomycete genomes and are direct targets of WhiAB (*Figure 1—figure supplement 1*). This analysis turned our attention to *vnz_27360* (here named *sepH* for 'septation protein H'), a gene of previously unknown biological function that is conserved across the *Streptomyces* genus. Bioinformatic analysis revealed that SepH consists of an N-terminal domain of unknown function (DUF3071) and an unstructured, less conserved C-terminal half (*Figure 1B*). In addition, the DUF3071 domain contains a predicted helix-turn-helix (HTH) motif, suggesting that SepH could function as a DNA-binding protein.

To investigate if *sepH* plays a role in *Streptomyces* cell division, we first generated a Δ*vnz_27360::apr* null mutant (Δ*sepH*) and imaged sporulating hyphae of wild-type (WT) *S. venezuelae*, Δ*sepH*, and the complemented Δ*sepH* mutant strain (Δ*sepH/sepH⁺*) by cryo-scanning electron microscopy (cryo-SEM) (*Figure 1C*). While aerial hyphae of WT *S. venezuelae* completely differentiated into chains of regularly sized spores, *sepH*-deficient hyphae failed to undergo efficient sporulation septation, leading to chains of spores of irregular size (*Figure 1—figure supplement 2*). This phenotype could be fully complemented by providing *sepH in trans* (Δ*sepH/sepH⁺*), confirming that SepH is indeed required for normal sporulation.

Next, we set out to determine the subcellular localization of SepH and generated a C-terminal SepH-YPet fusion. The *sepH-ypet* fusion, controlled by its native promoter, was integrated *in trans* at the ΦBT1 phage attachment site in a merodiploid strain that additionally expressed an ectopic copy of *mcherry* labeled *ftsZ,* which was under control of the native promoter. Microscopic analysis of the resulting *S. venezuelae* strain revealed that SepH-YPet co-localized specifically with FtsZ-mCherry at incipient division sites including vegetative cross-walls and sporulation septa (*Figure 1D*). Although *sepH* is a direct target of the two sporulation-specific regulators WhiAB, the

accumulation of SepH-YPet at vegetative division septa suggests that *sepH* expression is probably driven from an additional, WhiAB-independent promoter and that SepH might also be involved in cell division during vegetative growth.

Furthermore, we asked whether the specific localization pattern of SepH is dependent on FtsZ and the assembly of a functional divisome. To address this question, we took advantage of an Δ*ftsZ* null mutant (*Santos-Beneit et al., 2017*) and inserted a *sepH-ypet* fusion *in trans*. Fluorescence microscopy of the Δ*ftsZ* strain constitutively producing SepH-YPet revealed that in the absence of FtsZ, SepH-YPet was largely stable and mostly dispersed in the cytoplasm (*Figure 1—figure supplement 3A and B*), indicating that SepH recruitment to future division sites depends upon the assembly of Z-rings.

## SepH is important for cell division during vegetative growth and sporulation

To determine the role of SepH in *Streptomyces* cell division, we introduced a *ftsZ-ypet* gene fusion into the Δ*sepH* mutant strain and WT *S. venezuelae* and followed the formation of Z-rings during sporulation using time-lapse fluorescence microscopy. In sporulating WT hyphae, Z-rings are assembled in a characteristic, 'ladder-like' pattern in the tip cell compartment of sporogenic hyphae (*Schlimpert et al., 2016*; *Schwedock et al., 1997*). These so-called 'Z-ladders' lead to the synthesis of sporulation septa and the formation of chains of exospores of equal size (*Figure 2A* and *Video 1*). Interestingly, in the Δ*sepH* mutant Z-ladders were less uniform and frequently displayed an irregular spacing between individual FtsZ-YPet-rings. As observed in cryo-SEM images, spores produced under these conditions were aberrant in size and shape, indicating that regular septation was impaired (*Figure 2B* and *Video 2*).

Kymograph analyses of FtsZ-YPet fluorescence in sporulating WT hyphae confirmed the expected regular spacing and dynamics of Z-rings, including Z-ring assembly and constriction, which is accompanied by an increase in fluorescence followed by disassembly and loss of defined FtsZ-YPet fluorescence (*Figure 2C and E* and *Figure 2—figure supplement 1A*). In contrast, establishment of equally spaced Z-rings frequently failed in *sepH*-deficient hyphae leading to the formation of larger, spore-like compartments (*Figure 2D* and *Figure 2—figure supplement 1B*). Notably, closer inspection of the gaps within Z-ladders in Δ*sepH* mutant hyphae revealed that no discrete FtsZ-YPet signal was visible at these positions, indicating that the formation of individual Z-rings was disturbed very early in the assembly process. However, we found that the dynamics of Z-ring assembly, constriction and disassembly of the remaining Z-rings in *sepH*-deficient hyphae were very similar to the WT (*Figure 2E*). We also note that FtsZ protein levels were comparable in sporulating WT and Δ*sepH* cultures (*Figure 2—figure supplement 2*), indicating that the absence of SepH does not affect FtsZ protein stability. Moreover, calculation of the average Z-ring width in the two strains did not reveal any marked differences (*Figure 2F*), suggesting that the overall dynamics and architecture of Z-rings are WT-like in Δ*sepH* mutant hyphae and that additional mechanisms might be in place that can partially compensate for the lack of SepH activity.

While analyzing the spatiotemporal localization of FtsZ-YPet in Δ*sepH* hyphae, we noticed occasional lysis of large hyphal segments and the formation of unusual branched sporogenic hyphae (*Figure 2B*). Given that FtsZ and SepH also co-localize at cross-walls (*Figure 1D*), we reasoned that the absence of SepH might also affect cell division during vegetative growth. To examine the importance of SepH for cross-wall formation, we used the fluorescent D-ala-D-ala analogue HADA to label peptidoglycan and to visualize cross-walls (*Kuru et al., 2015*). Inspection of still images of WT *S. venezuelae* and the complemented Δ*sepH* mutant strain (Δ*sepH/sepH*⁺) grown in the presence of HADA showed comparable frequency and distribution of cross-walls within vegetative hyphae. However, we found that *sepH*-deficient hyphae displayed visibly fewer cross-walls compared to the WT (*Figure 2G*). The dramatically reduced number of cross-walls in the *sepH* mutant could explain the lysis and branching phenotype we observed in our Δ*sepH* time-lapse microscopy experiments. In WT *Streptomyces*, cross-walls compartmentalize growing hyphae and are often associated with hyphal branch points leading to the physical separation of different hyphal segments. Thus, fewer cross-walls in the Δ*sepH* mutant result in much longer hyphal compartments that are potentially more susceptible to large-scale lytic events. In addition, at the onset of sporulation septation, FtsZ ladders assemble within these unsegmented and branching hyphal compartments which can subsequently result in the formation of the enlarged triangular shaped-like spores at hyphal branch points

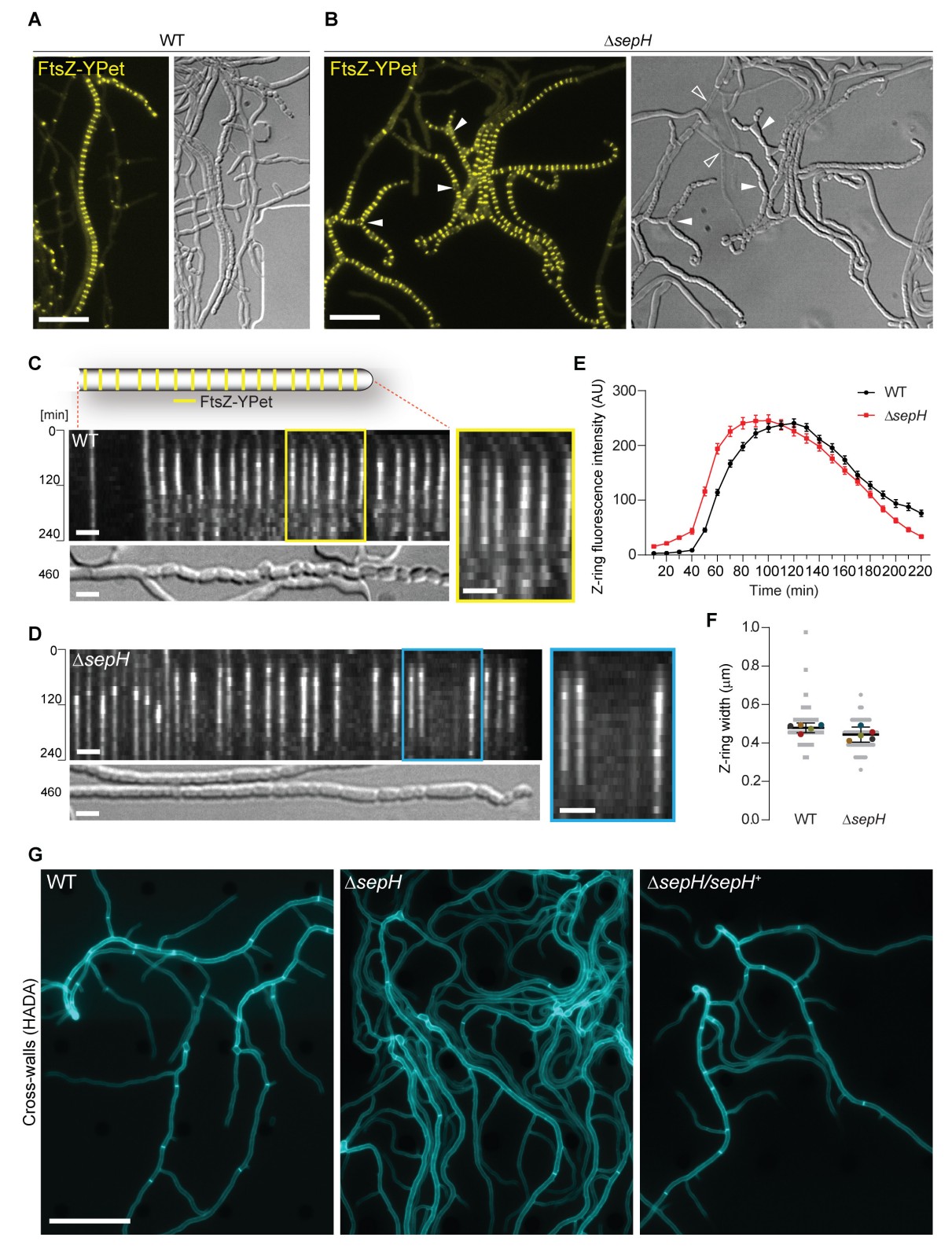

**Figure 2.** SepH is important for cell division leading to sporulation septa and vegetative cross-walls. (A and B) Still images from *Videos 1* and *2* showing the localization of FtsZ-YPet in sporulating (A) wild-type (WT; SS12) and (B) Δ*sepH* mutant hyphae (MB750). Arrow heads in (B) point at aberrant spores or gaps in FtsZ-YPet-ladders (filled arrow head) or indicate lysed hyphae (open arrow heads in DIC image). Note that DIC images correspond to a later time point than fluorescence micrographs to show the terminal sporulation phenotype. Scale bars: 10 µm. (C) and (D) Kymograph
*Figure 2 continued on next page*

*Figure 2 continued*

analysis of FtsZ-YPet dynamics during sporulation-specific cell division in WT (**C**) and Δ*sepH* hyphae (**D**), ectopically expressing an additional copy of *ftsZ-ypet* from the native promoter (SS12 and MB750). The DIC image below shows the terminal sporulation phenotype. Yellow and blue boxes indicate magnified regions of the kymograph. Scale bar: 2 μm. Additional examples of kymographs can be found in *Figure 2—figure supplement 1*. (**E**) Fluorescence intensity traces of FtsZ-YPet (Z-rings) over time derived from sporulating WT (SS12) and Δ*sepH* mutant hyphae (MB750). Shown are the mean fluorescence intensity traces (mean ± SEM) collected from Z-rings of five sporulating hyphae for each strain. (**F**) Width of Z-rings in sporulating hyphae of WT (SS12) and *sepH*-deficient hyphae (MB750). The same data set as in (**E**) was used and the mean width for reach replicate (colored dots, n = 5) ±95% CI was plotted. (**G**) HADA labeling of peptidoglycan to visualize cross-walls in WT, Δ*sepH* (SV56), and Δ*sepH*/*sepH*⁺ (MB747) hyphae. Spores of each strain were germinated and grown in the presence of 0.25 mM HADA for 5 hr in a microfluidic device before imaging. Scale bar: 20 μm.

The online version of this article includes the following source data and figure supplement(s) for figure 2:

**Source data 1.** Time-lapse fluorescence image series of selected and straightened hyphae (SS12) used to generate kymograph shown in *Figure 2C*.
**Source data 2.** Time-lapse fluorescence image series of selected and straightened hyphae (MB750) used to generate kymograph shown in *Figure 2D*.
**Source data 3.** Custom Python and R-scripts and extracted fluorescence intensities of Z-rings used to generate *Figure 2E and F*.
**Figure supplement 1.** Additional examples of kymographs showing the localization of FtsZ-YPet over time in (**A**) wild-type (SS12) or (**B**) Δ*sepH* hyphae (MB750).
**Figure supplement 2.** FtsZ levels in wild-type and Δ*sepH* cells during sporulation.

(*Figure 2B*). Collectively, these results demonstrate a crucial role for SepH in cell division during vegetative growth and sporulation.

## The N-terminal DUF3071 domain is crucial for SepH function

To identify the protein regions required for the recruitment and function of SepH, we generated fluorescent protein fusions to the N-terminal DUF3071 domain (SepH-NTD, residues 1–186) and the unstructured C-terminal domain (SepH-CTD, residues 187–344) (*Figure 3A*). The corresponding mutant alleles were integrated *in trans* at the ΦBT1 phage attachment site in the Δ*sepH* mutant and expressed from the native promoter. The resulting strains were then analyzed by fluorescence microscopy and cryo-SEM to determine the subcellular localization of the fusion proteins and their ability to compensate for the loss of WT SepH activity. Using automated Western blotting, we verified that all proteins were synthesized and stable under the conditions used (*Figure 3—figure supplement 1*). Control experiments with full-length SepH-YPet demonstrated that this fusion was fully functional and restored WT-like localization and sporulation (compare *Figure 3B and F* and *Figure 3—figure supplement 2*). Expression of the SepH C-terminal domain (SepH-CTD) gave a diffuse localization pattern and failed to complement any aspect of the Δ*sepH* phenotype, resulting in irregular sporulation and cell lysis (*Figure 3C and G*). Interestingly, production of just the N-terminal DUF3071 domain (SepH-NTD) was sufficient to partially restore normal sporulation (*Figure 3H*). However, the distinct septal

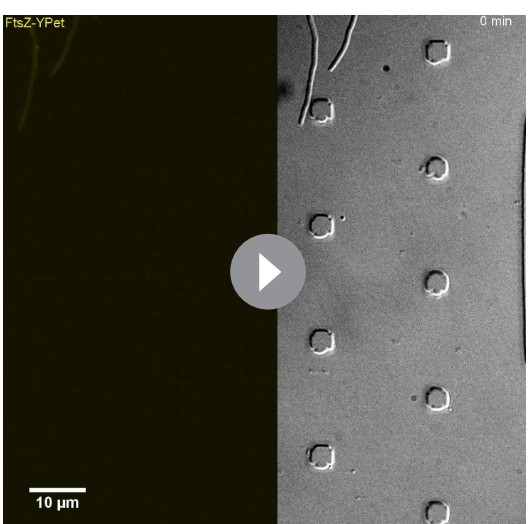

**Video 1.** Time-lapse fluorescence and DIC microscopy movie showing the localization of FtsZ-YPet in vegetative and sporulating hyphae of wild-type *S. venezuelae* (SS12). Scale bar: 10 μm.
https://elifesciences.org/articles/63387#video1

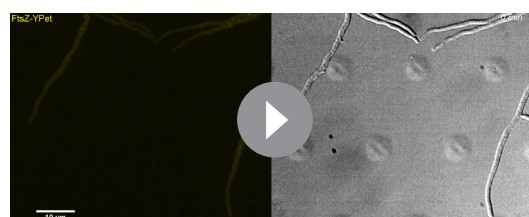

**Video 2.** Time-lapse fluorescence and DIC microscopy movie showing growth and localization of FtsZ-YPet in vegetative and sporulating hyphae of the *S. venezuelae* Δ*sepH* mutant (MB750). Scale bar: 10 μm.
https://elifesciences.org/articles/63387#video2

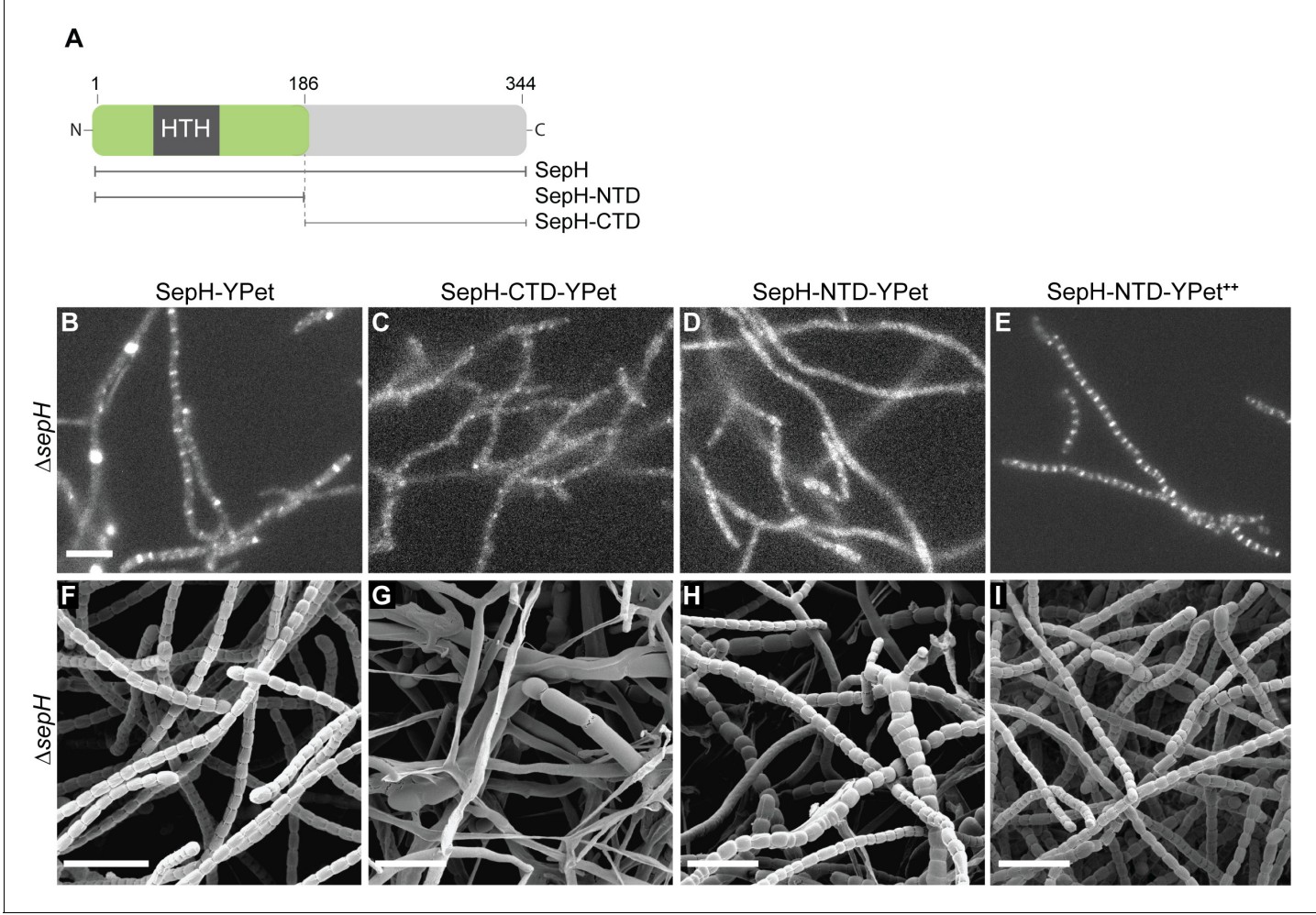

**Figure 3.** The DUF3071 domain is crucial for SepH function *in vivo*. (**A**) Schematic showing the SepH domain architecture and constructed truncations. Numbers indicate the relevant amino acid positions. (**B–E**) Fluorescence micrographs showing the localization of the full-length and truncated SepH-YPet variants in the Δ*sepH* mutant expressed from the native promoter (**B–D**, strains MB918, MB827, MB828) or from the constitutive $P_{ermE*}$ promotor (**E**, strain MB852). Scale bar: 5 μm. (**F–I**) Cryo-SEM images of the same strains presented in (**B–E**) showing the ability of (**F**) full-length SepH-YPet or (**G–I**) truncated versions of SepH fused to YPet to complement the sporulation defect of the Δ*sepH* mutant when produced from the native promoter (**F–H**, strains MB918, MB827, MB828) or a constitutive promoter (**I**, strain MB852). Note, expression of *sepH-CTD* (**G**) does not rescue the Δ*sepH* mutant. Scale bars: 5 μm.

The online version of this article includes the following figure supplement(s) for figure 3:

**Figure supplement 1.** Automated Western blot analysis of the different SepH-YPet constructs shown in *Figure 3B–D*.

**Figure supplement 2.** Control image showing the Δ*sepH* phenotype.

**Figure supplement 3.** Constitutive expression of *sepH-CTD-ypet* does not rescue the Δ*sepH* phenotype.

accumulation characteristic of full-length SepH-YPet and WT-like sporulation could only be clearly observed for the truncated gene fusion when it was expressed from a constitutive promoter (*ermE*$*_p$) in the Δ*sepH* mutant (*Figure 3E and I*). By contrast, constitutive expression of *sepH-CTD-ypet* did not improve sporulation in the Δ*sepH* mutant (*Figure 3—figure supplement 3*). Taken together, these results imply that the conserved N-terminal region of SepH encoding the DUF3071 domain is vital for SepH function, but WT activity also requires the C-terminal domain.

## SepH does not bind to the nucleoid

The DUF3071 domain of SepH includes a HTH motif, characteristic of DNA binding proteins (*Aravind et al., 2005*). This raised the question as to whether SepH could interact with the nucleoid. Notably, to-date, no functional homologs of the well-described nucleoid occlusion systems present

in other bacteria have been identified in *Streptomyces*. To investigate a potential role of SepH in chromosome segregation, we first generated a dual-labeled strain which produced SepH-YPet and a mCherry-labeled version of the bacterial nucleoid-associated protein HupA (*Salerno et al., 2009*). Both fluorescent protein gene fusions were integrated at the ΦBT1 phage attachment site of WT *S. venezuelae* and expression was driven from their native promoters. Fluorescence microscopy of the resulting *S. venezuelae* strain showed that SepH-YPet and HupA-mCherry did not co-localize. Instead, SepH-YPet accumulated at sites where HupA-mCherry was largely absent, indicating that SepH does not associate with the nucleoid (*Figure 4A*). Furthermore, we visualized the nucleoid in WT and Δ*sepH* spore chains stained with the fluorescent DNA dye 7-AAD but did not observe anucleate spores in the Δ*sepH* mutant (*Figure 4B*), suggesting that chromosome segregation is not impaired in *sepH*-deficient hyphae.

To independently verify our localization studies, we first performed chromatin immunoprecipitation coupled with deep-sequencing (ChIP-seq) using sporulating WT *S. venezuelae*. In parallel, we conducted ChIP-seq experiments with the Δ*sepH* mutant strain as a negative control to eliminate false-positive signals arising from non-specific binding of the α-SepH antibody. Analysis of the ChIP-seq results did not reveal any significant enrichment of SepH on the chromosome (*Figure 4—figure supplement 1A*) compared to the Δ*sepH* negative control. Furthermore, DNase I footprinting experiments using purified SepH together with radiolabeled probes derived from three of the most enriched chromosomal regions (*Figure 4—figure supplement 1B*) did not reveal protection of the selected DNA fragments, collectively suggesting that SepH does not bind to specific DNA-sequences. Finally, we performed electrophoretic mobility shift assays (EMSAs) to test SepH for non-specific DNA-binding activity *in vitro* (*Figure 4—figure supplement 1C*). For this, we tested binding of SepH to the promoter region of *vnz_35870* (the most enriched region in ChIP-seq), the sequence internal to *vnz_08520* (not enriched in ChIP-seq) and the low GC-sequence of the kanamycin resistance gene from a standard *E. coli* expression vector. Under the conditions used, we did not observe binding of SepH to any of these DNA fragments. Collectively, our results strongly suggest that the HTH motif in the conserved N-terminal region of SepH is not involved in DNA binding and that SepH does not play a direct role in chromosome segregation.

## The SepH HTH motif is essential for the interaction with FtsZ

While most HTH motifs mediate DNA binding, exceptions to this rule exist and HTH motifs have also been shown to facilitate protein–protein interaction (*van den Ent et al., 2010*). Thus, we hypothesized that the HTH motif within the NTD of SepH could directly affect the function of a protein binding partner, such as FtsZ or other components of the *Streptomyces* cell division machinery. To investigate this possibility, we performed yeast-two hybrid (Y2H) assays. As previously described, we observed that FtsZ can self-interact and associate with SepF (*Schlimpert et al., 2017*). Furthermore, our Y2H experiments suggested that SepH can oligomerize and, most significantly, SepH binds FtsZ (*Figure 5A*). In addition, we tested interactions between SepH and several other *Streptomyces* divisome components including SepF, SepF2, SepF3, DynA, and DynB but only detected a putative interaction with SepF in one orientation (*Figure 5—figure supplement 1*).

To identify the SepH domain involved in binding FtsZ, we performed additional experiments using the SepH-NTD and the SepH-CTD variants (*Figure 5B*). We found that the SepH-NTD could bind FtsZ in the Y2H assays but the SepH-CTD could not. We hypothesized that the HTH fold in the N-terminal domain of SepH could be involved in binding FtsZ. Thus, we repeated the Y2H assay with a SepH variant in which we had substituted a highly conserved glycine residue in the HTH motif with a proline residue (SepH-G79P) (*Mercy et al., 2019*). While SepH-G79P was still able to interact with WT SepH (*Figure 5B*), this mutant version failed to bind FtsZ. This indicated that the SepH HTH motif is indeed required for the interaction with FtsZ, but not for self-interaction. We also introduced the *sepH*-G79P allele into the Δ*sepH* mutant and found that SepH-G79P was unable to restore WT-like sporulation (*Figure 5—figure supplement 2*).

Encouraged by the Y2H results, we purified recombinant *S. venezuelae* FtsZ, SepH, and the SepH variants to test if SepH can directly influence the behavior of FtsZ *in vitro* (*Figure 5C*). Using circular dichroism (CD), we first confirmed that the G79P substitution did not cause any major structural changes (*Figure 5—figure supplement 3A*). We then examined SepH and its variants by size exclusion chromatography and found that WT SepH, SepH-CTD, and SepH-G79P eluted as one peak, corresponding to a predicted size of a tetramer, while SepH-NTD eluted as a dimer (*Figure 5—figure*

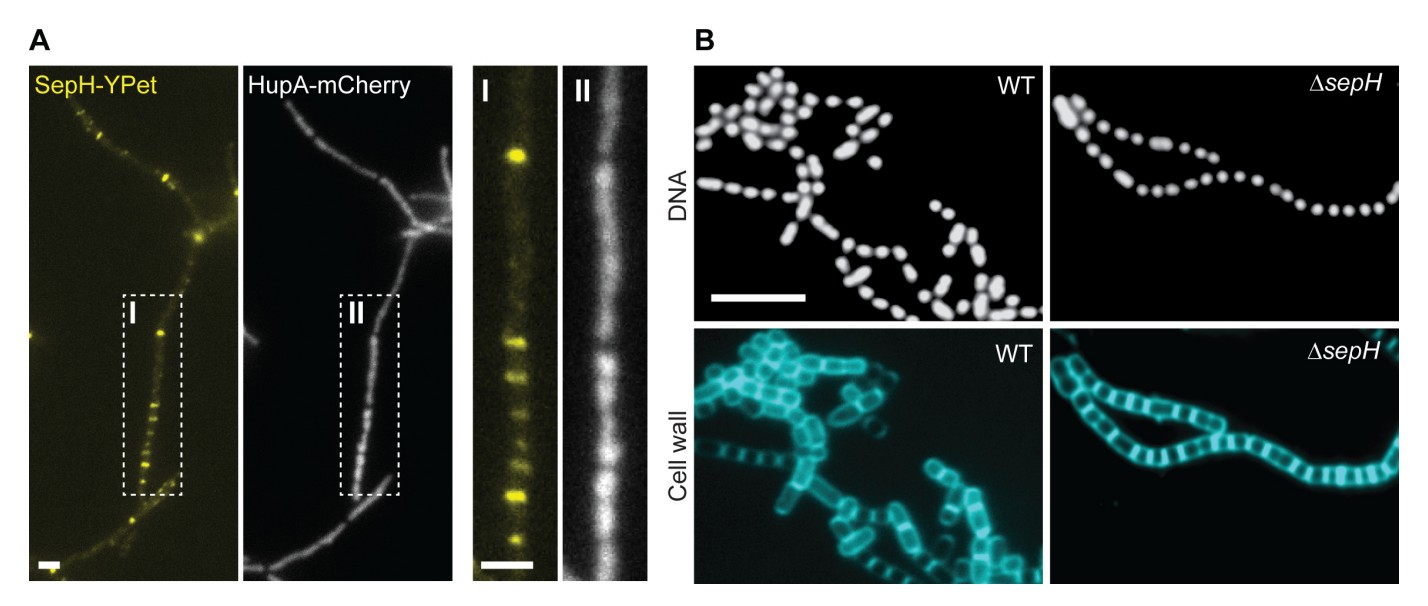

**Figure 4.** SepH is not associated with the nucleoid or required for chromosome segregation. (**A**) Fluorescence micrographs showing the accumulation of SepH-YPet and the concomitant distribution of chromosomal DNA visualized using the nucleoid-associated protein HupA fused to mCherry (MB807). Boxes I and II indicate an enlarged hyphal segment shown in the left panels. Scale bars: 2 µm. (**B**) Fluorescence images of wild-type (WT) and Δ*sepH* (SV56) spore chains incubated with the fluorescent dyes 7-AAD and WGA Alexa Fluor 488 to visualize DNA and cell wall material, respectively. Scale bar: 5 µm.

The online version of this article includes the following figure supplement(s) for figure 4:

**Figure supplement 1.** SepH does not bind DNA.

*supplement 3B*). We next measured the GTP hydrolysis rate of FtsZ in the presence of SepH (*Figure 5D*). Initial characterization of the biochemical properties of *S. venezuelae* FtsZ confirmed that the protein was active and robustly hydrolyzed GTP in a time-dependent manner at a rate of 1.12 ± 0.44 GTP per FtsZ per minute, similar to the recently published activity of *Streptomyces* FtsZ (*Figure 5—figure supplement 3C*; *Sen et al., 2019*). While SepH on its own did not hydrolyze GTP, we found that the addition of increasing amounts of SepH led to a moderate increase of FtsZ's GTP turnover (*Figure 5D*). Importantly, we measured a similar increase in GTP turnover when we incubated FtsZ with SepH-NTD but not with the SepH-CTD or SepH-G79P variants (*Figure 5E*).

To ensure that our studies on the effect of SepH on FtsZ behavior were performed under physiologically relevant conditions, we used quantitative Western blot analysis and determined that the intracellular levels of SepH are sixfold lower than FtsZ (*Figure 5—figure supplement 4*). Notably, we found in our initial GTPase assays that using 0.6 µM SepH was already sufficient to stimulate the GTPase activity of 3.5 µM FtsZ (*Figure 5D*). Based on these results, we performed all subsequent *in vitro* studies using a 1:6 molar ratio of SepH to FtsZ.

To further substantiate our finding that SepH directly binds FtsZ, we performed high-speed sedimentation assays (*Figure 5F*). In the absence of GTP, FtsZ was unable to polymerize into filaments and was largely found in the supernatant after ultracentrifugation. In the presence of GTP, approximately 35% of FtsZ was detected in the pellet fraction, indicating that FtsZ had assembled into polymers. When SepH was added to the reaction with FtsZ and GTP prior to ultracentrifugation, 47% of FtsZ and 96% of SepH were robustly pelleted, confirming a direct interaction between SepH and FtsZ. In contrast, incubation of FtsZ with GTP and the SepH HTH mutant variant (SepH-G79P) resulted in a clearly reduced co-sedimentation of SepH-G79P (50%) with polymerized FtsZ (36%). In addition, in the absence of FtsZ both SepH and SepH-G79P were largely soluble.

Next, we asked whether the hydrolysis of GTP is required for the interaction of SepH with FtsZ. To address this question, we repeated the co-sedimentation assays with FtsZ and SepH using the slow-hydrolysable GTP analogue GMPCCP. Under these conditions, the amount of FtsZ in the pellet fraction nearly doubled (88%) and this was independent of SepH, which co-sedimented almost

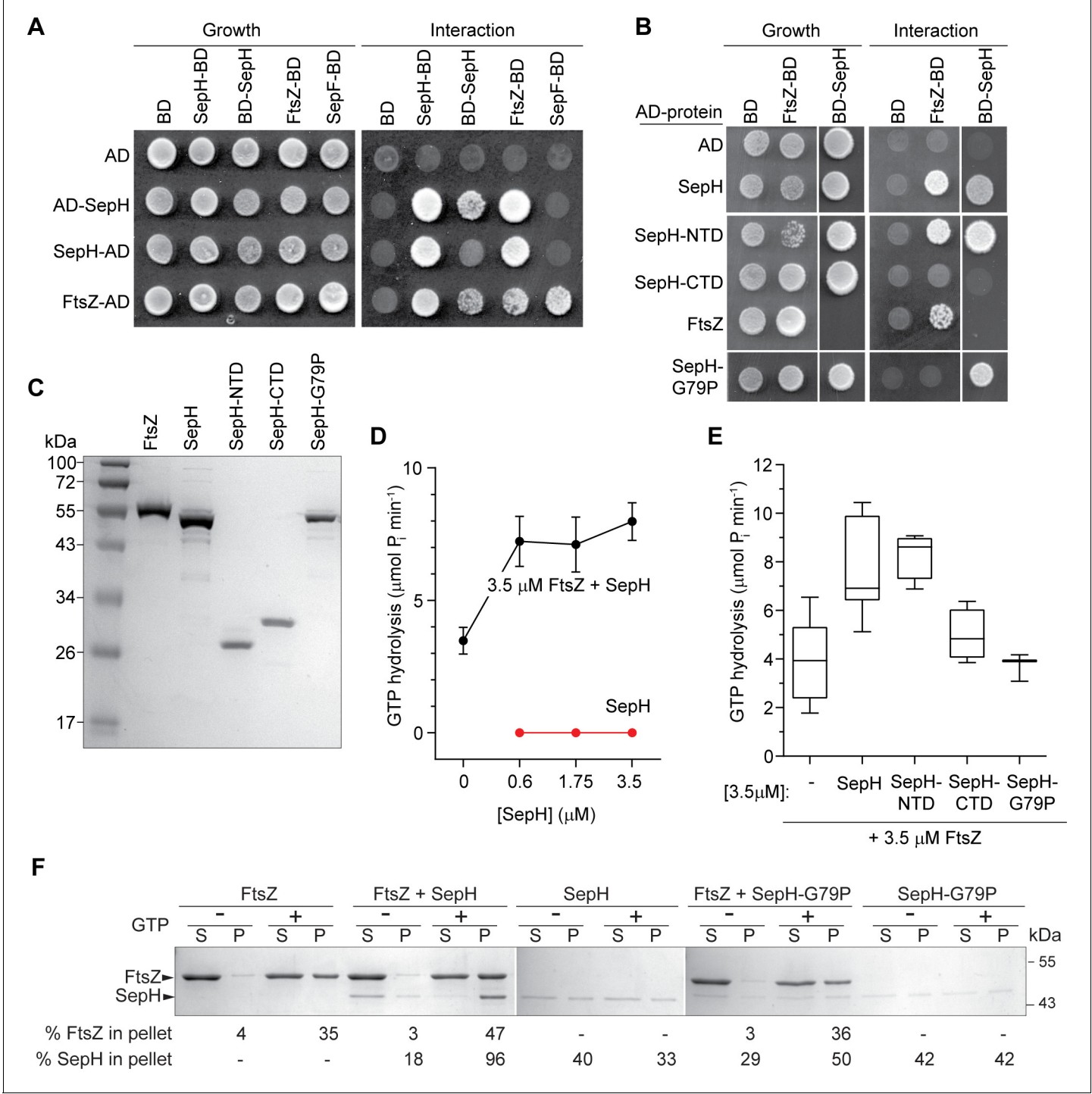

**Figure 5.** SepH helix-turn-helix (HTH) motif is crucial for the interaction with FtsZ. (**A**) Yeast two-hybrid analysis. The indicated proteins were fused to the GAL4 activation domain (AD) and the GAL4 DNA-binding domain (BD). The viability of the yeast strains expressing the respective fusion proteins was confirmed by spotting the individual strains on minimal medium containing leucine and tryptophan (left panel). Interaction between the protein fusion allows growth on minimal medium lacking leucine, tryptophan, histidine, and alanine (right panel). The full set of tested interactions can be found in *Figure 5—figure supplement 1*. Each interaction was tested in triplicate. (**B**) Yeast two-hybrid assay showing the interaction between FtsZ and different SepH variants, including full-length SepH (SepH), the N-terminal domain of SepH (SepH-NTD), the C-terminal SepH domain (SepH-CTD), and the mutated SepH HTH domain (SepH-G79P). Experiments were performed as described above. (**C**) Coomassie-stained SDS gel with purified *S. venezuelae* FtsZ, SepH, SepH-NTD (residues 1–186), SepH-CTD (residues 187–344), and SepH-G79P. (**D**) Mean GTP hydrolysis rate of 3.5 µM FtsZ alone or in the presence of increasing concentrations of SepH. SepH did not show GTPase activity (red graph). Error bars represent SEM (n ≥ 3). (**E**) Mean

*Figure 5 continued on next page*

*Figure 5 continued*

GTP hydrolysis rate of FtsZ (3.5 µM) in the presence of wild-type SepH and SepH variants at a molar ratio of 1:1. Error bars represent min/max values (n ≥ 3). (**F**) Co-sedimentation of SepH with polymerized FtsZ *in vitro*. 3.5 µM FtsZ was incubated for 15 min in the presence or absence of 2 mM GTP, and 0.6 µM SepH or SepH-G79P as indicated. Polymerized FtsZ was collected by high-speed ultracentrifugation. The presence of proteins in the supernatant (S) and pellet (P) was analyzed by SDS-PAGE and Coomassie staining. Representative images of two independent experiments are shown. The percentage of total FtsZ or SepH/SepHG79P in the pellet fraction is indicated below. Note that due to number of samples, protein fractions were loaded on separate protein gels which resulted in slightly different staining intensity.

The online version of this article includes the following source data and figure supplement(s) for figure 5:

**Source data 1.** High-speed co-sedimentation data used in *Figure 5F*.
**Figure supplement 1.** Yeast-two hybrid analysis showing the complete set of tested interactions between SepH and different cell division proteins.
**Figure supplement 2.** Cryo-SEM micrograph of sporulating Δ*sepH* hyphae expressing *sepH-G79P* ectopically from the native promoter (MB938).
**Figure supplement 3.** Biochemical characterization of SepH, SepH variants, and FtsZ.
**Figure supplement 3—source data 1.** Analytical gel filtration data.
**Figure supplement 4.** Quantification of SepH and FtsZ abundance by quantitative (automated) Western blotting.
**Figure supplement 4—source data 1.** High-speed co-sedimentation data with GMPCPP.
**Figure supplement 5.** High-speed co-sedimentation with GMPCPP.

completely with FtsZ (98%) (*Figure 5—figure supplement 5*). Taken together, our two-hybrid and *in vitro* experiments demonstrate that SepH directly interacts with FtsZ via the HTH motif in the conserved N-terminal DUF3071 domain and that this interaction is independent of the GTPase activity of FtsZ.

## SepH stimulates the formation of dynamic FtsZ filaments *in vitro*

The enrichment of SepH and FtsZ in the pellet fraction following high-speed centrifugation suggested that SepH either promotes the formation of macromolecular FtsZ bundles or stimulates the assembly of a high number of individual FtsZ protofilaments. To distinguish between these two possibilities, we first repeated the co-sedimentation assays with GTP at a lower centrifugation speed, which would only allow the pelleting of FtsZ bundles but not FtsZ protofilaments. A similar approach was recently employed to examine the assembly state of FtsZ filaments in complex with the FtsZ-stabilizing protein GpsB (*Eswara et al., 2018*). Using this differential centrifugation method, we detected no meaningful accumulation of SepH (12%) and FtsZ (3%) in the pellet fraction (*Figure 6—figure supplement 1*), indicating the absence of large FtsZ assemblies in the presence of SepH.

To further investigate the assembly state of FtsZ and to visualize the effect of SepH on FtsZ filament morphology, we used negative staining and transmission electron microscopy (TEM). Our control experiments confirmed that purified FtsZ and SepH did not form any visible complexes when imaged on their own (*Figure 6A and B*). Upon addition of 2 mM GTP to the polymerization buffer (50 mM HEPES pH 7.2, 50 mM KCl, 5 mM MgCl₂), FtsZ (3.5 µM) formed long, gently curved fibers that sparsely covered the EM grid (*Figure 6C*). In the presence of GTP and 0.6 µM SepH, FtsZ filaments became readily visible and were similar in morphology but varied more in length compared to FtsZ filaments assembled without SepH (*Figure 6D*). We also tested if the addition of a higher concentration of SepH would affect FtsZ filament morphology. At an equimolar ratio of SepH and FtsZ (1:1), FtsZ filaments were highly abundant on EM grids and discernible as largely straight filaments of various lengths (*Figure 6—figure supplement 2A*). Occasionally, we did observe some thin filament bundles which were likely a result of the artificial stabilization of FtsZ protofilaments due to the excess of SepH rather than active stabilization of lateral interactions between FtsZ filaments. In addition, we examined FtsZ polymers in the presence of the different SepH variants. As expected, incubation of FtsZ with GTP and SepH-CTD or SepH-G79P did not alter FtsZ filament morphology (*Figure 6—figure supplement 2B-C*). However, FtsZ filaments appeared to be more abundant and longer in the presence of SepH-NTD (*Figure 6—figure supplement 2D*). Likewise, stable FtsZ filaments formed in the presence of slow-hydrolysable GTP analogue GMPCPP either with or without SepH were similar in appearance and did not assemble into large polymer aggregates (*Figure 6—figure supplement 2E and F*). We therefore conclude that SepH does not actively contribute to the bundling of FtsZ protofilaments *in vitro*.

To directly follow the assembly kinetics of purified FtsZ into filaments, we used dynamic light scattering (DLS). In the presence of 50 µM GTP, FtsZ monomers assembled into protofilaments, which

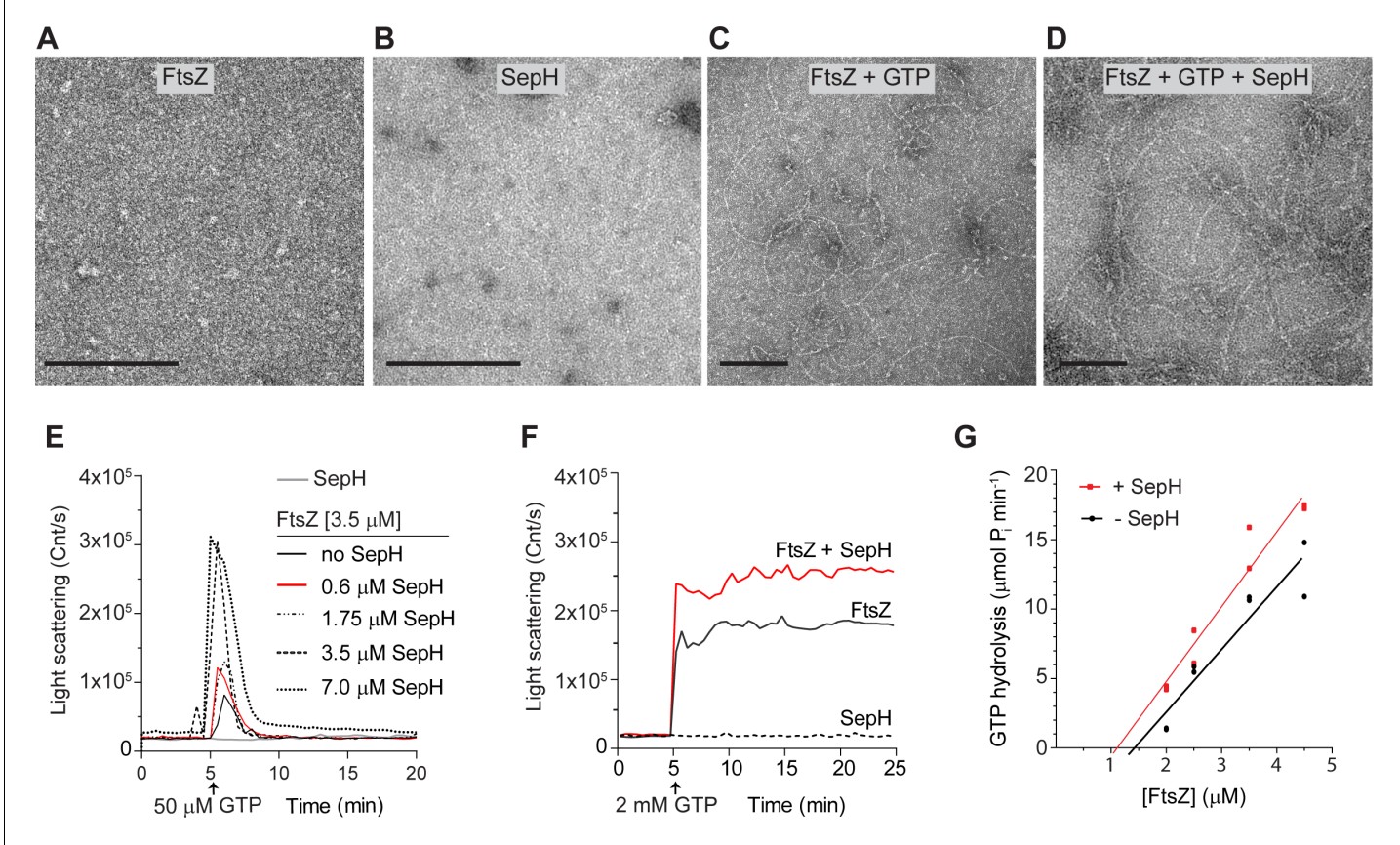

**Figure 6.** SepH stimulates the polymerization of dynamic FtsZ protofilaments. (**A–D**) Visualization of purified FtsZ and/or SepH using negative staining transmission electron microscopy (TEM). No structures were detected for 3.5 μM FtsZ in the absence of GTP (**A**), or 0.6 μM SepH in the presence of GTP (**B**). Filaments were observed for FtsZ (3.5 μM) when 2 mM GTP was added (**C**), and increased FtsZ polymerization was observed when SepH (0.6 μM) was added to the reaction (**D**). Scale bar: 100 nm. (**E**) Light scatter traces showing the reversible assembly of 3.5 μM FtsZ filaments in the presence of 50 μM GTP and increasing amounts of SepH. Red line denotes DLS trace generated with a molar ratio of FtsZ to SepH at 6:1. (**F**) Light scatter traces showing the polymerization of 3.5 μM FtsZ with 2 mM GTP in the presence (red line) or absence of 0.6 μM SepH (black line). SepH alone did not generate light scattering when incubated with 2 mM GTP (dashed line). Light scatter graphs display representative traces of at least three independent experiments. (**G**) Critical concentration (Cc) of FtsZ from *S. venezuelae* in the presence and absence of 0.6 μM SepH. Cc was determined by extrapolating the linear regression line of the FtsZ GTPase rate backwards to where it intercepts the X-axis. GTPase hydrolysis rates are the result of two independent experiments.

The online version of this article includes the following source data and figure supplement(s) for figure 6:

**Figure supplement 1.** Low-speed co-sedimentation with GTP.
**Figure supplement 1—source data 1.** Low-speed co-sedimentation data with GTP.
**Figure supplement 2.** FtsZ filament morphology in the presence of excess SepH, SepH variants, and with GMPCCP.
**Figure supplement 3.** Additional FtsZ polymerization results as measured by DLS.

resulted in a sharp increase in the light scattering signal. The reaction reached a brief steady-state level before GTP became limiting and the intrinsic FtsZ GTPase activity triggered depolymerization and the complete disassembly of FtsZ filaments. Importantly, incubation of 3.5 μM FtsZ with 50 μM GTP and increasing amounts of SepH led to a significantly higher amplitude in light scattering compared to FtsZ with just GTP (*Figure 6E*). The initial burst in light scattering was followed by a rapid decrease in the light scatter signal and the complete depolymerization of FtsZ filaments. Control experiments using GDP or SepH with GTP did not generate a light scattering signal, confirming the absence of any polymers or molecular assemblies (*Figure 6—figure supplement 3A*).

We repeated the DLS experiments using a higher GTP concentration (2 mM) and a ratio of FtsZ and SepH of 6:1 to test if the detected decrease in light scattering was caused by the depletion of GTP and the accumulation of GDP. When hydrolysable GTP was provided in excess, there was no

drop in the light scatter signal (*Figure 6F*), suggesting that the observed decline in the scatter signal in *Figure 6E* was in fact caused by the depolymerization of FtsZ filaments. Furthermore, we recorded the polymerization dynamics of FtsZ in combination with the different SepH variants. In line with our earlier results, SepH-NTD stimulated FtsZ assembly dynamics although not to the same degree as full-length SepH (*Figure 6—figure supplement 3B*). In contrast, SepH-CTD and SepH-G79P did not further affect the assembly kinetics of FtsZ and resulted in scatter profiles similar to FtsZ with just GTP. In addition, we monitored FtsZ polymerization using GMPCCP (*Figure 6—figure supplement 3C*). In agreement with the TEM results, we did not observe any change in the light scattering curves when SepH was added to the polymerization reaction, indicating the absence of higher molecular FtsZ assemblies (*Figure 6—figure supplement 2E and F*).

The DLS curves of FtsZ filament assembly with and without SepH showed a similar initial rate of polymerization but reached a higher light scatter amplitude when SepH was present (*Figure 6F*). We reasoned that SepH has a positive effect on FtsZ polymerization and that the associated increase in FtsZ GTPase activity is a direct consequence of a higher amount of FtsZ protofilament ends that can undergo treadmilling. To support this idea, we determined the critical concentration of FtsZ and found that the addition of SepH, at a molar ratio of 6 FtsZ to 1 SepH, lowers the critical concentration of FtsZ from 1.43 to 1.11 µM (*Figure 6G*). Together, these findings demonstrate that SepH directly regulates the behavior of FtsZ by promoting the reversible assembly of FtsZ protofilaments.

## SepH is conserved in morphologically diverse actinobacteria

Previous work by *Gao et al., 2006* identified a group of 24 so-called signature proteins that are highly conserved actinobacterial proteins and inculde SepH. To get a better understanding of the phylogenetic distribution and conservation of SepH, we specifically searched for SepH homologs in an expanded set of 3962 representative genomes, including those of 673 actinobacterial species. In total, we identified 626 SepH homologs, which, in agreement with Gao et al., are exclusively found in actinobacteria (*Figure 7A*; *Gao et al., 2006*). Furthermore, SepH homologs cluster into distinct groups, suggesting a greater sequence divergence at the family level. Interestingly, in contrast to SepH homologs detected in, for example, the Corynebacteriales or Micrococcales, SepH homologs identified in all analyzed streptomycetes genomes (n = 60) display a very high sequence identity (>80%), which is reflected by the small number of individual leaves within the Streptomycetales branch. Notably, members of the actinobacteria display remarkably diverse cellular morphologies, ranging from cocci and rods to multicellular filaments (*Barka et al., 2016*). Thus, it is conceivable that SepH homologs have further evolved to support cell division in the different actinobacterial species.

Despite this apparent divergence of SepH homologs throughout the actinobacteria, a refined alignment of 360 representative SepH sequences clearly showed a strong conservation in the N-terminal DUF3071 domain, including the HTH motif (*Figure 7—figure supplement 1*). Interestingly, we identified two additional highly conserved sequence motifs at the far C-terminal end of SepH homologs. These two motifs include a four-amino acid lysine and arginine-rich patch, which is particularly enriched in SepH sequences from *Corynebacteria*, and an additional string of 10 amino acids which is present in all analyzed SepH homologs (*Figure 7—figure supplement 1*). It is conceivable that these residues are involved in a yet unidentified aspect of SepH function.

To investigate if SepH homologs share a similar biological function, we expressed codon-optimized *sepH* from the non-pathogenic, rod-shaped model organism *Mycobacterium smegmatis* mc$^2$ 155 (*sepH$_{Ms}$, MSMEG_5685*) in the *S. venezuelae* Δ*sepH* mutant. Both SepH homologs share an overall sequence identity of 34%. The heterologous *sepH$_{Ms}$* was fused to *mcherry*, placed under the control of the native *sepH$_{Sv}$* promoter and integrated at the *S. venezuelae* ΦBT1 phage attachment site. SepH$_{Ms}$-mCherry was able to fully support WT-like sporulation in the Δ*sepH* mutant (*Figure 7B*). Moreover, SepH$_{Ms}$-mCherry also displayed the characteristic septal localization in growing and sporulating hyphae similar to SepH from *S. venezuelae* (*Figures 1D* and *7B*). In addition, we could detect a direct interaction between SepH$_{Ms}$ and FtsZ$_{Ms}$ using yeast two-hybrid analyses, supporting the idea that SepH plays a universal role in actinobacterial cell division (*Figure 7C*).

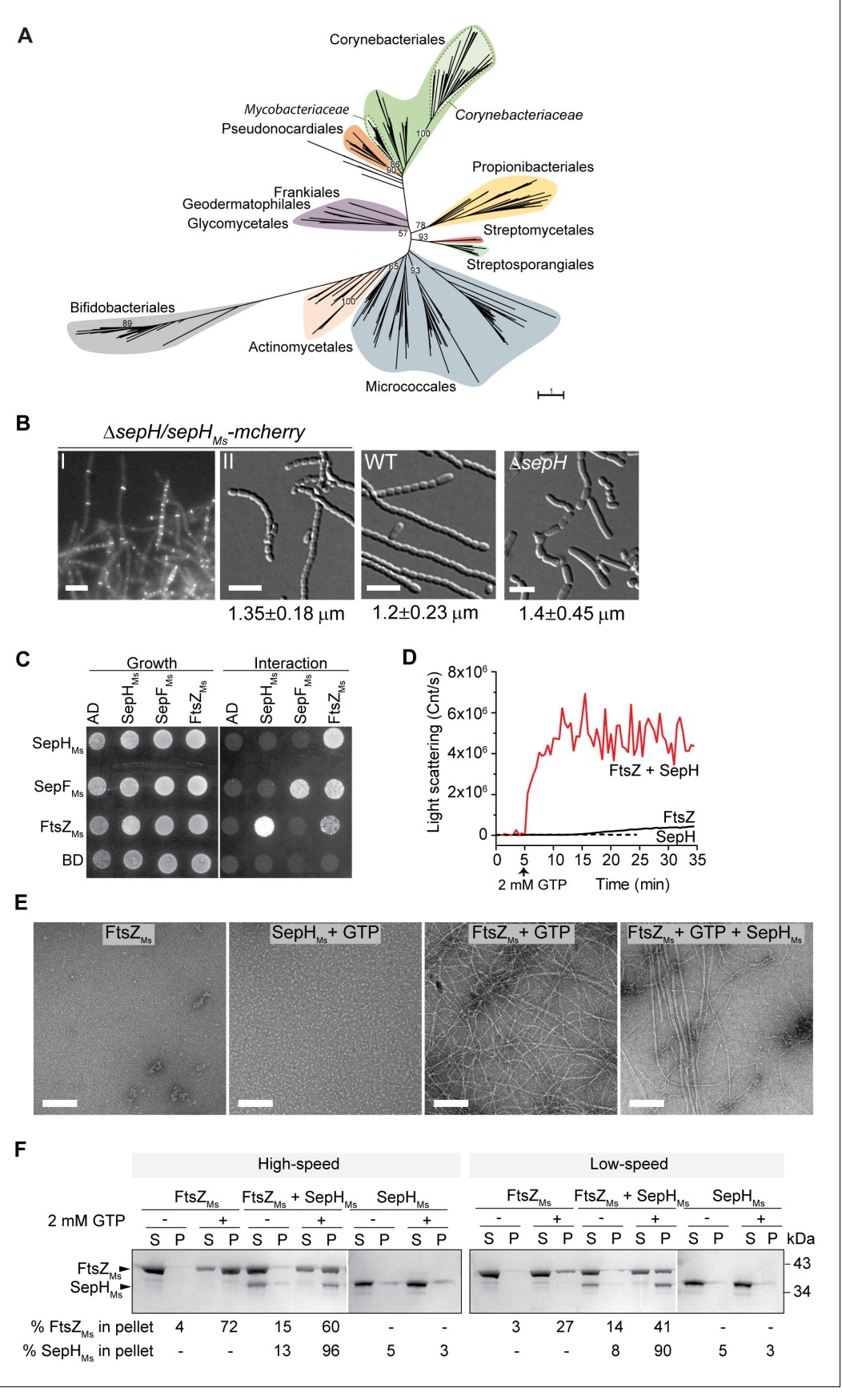

**A**

**B** ΔsepH/sepH_Ms-mcherry

**C**

**D**

**E**

**F**

**Figure 7.** SepH$_{Ms}$ stimulates FtsZ polymerization and bundling *in vitro*. (**A**) Phylogenetic tree showing the distribution of SepH within different actinobacterial orders. Major orders with more than two representative leaves are shown in different colors. Numbers denote bootstrap values. The scale bar represents the average substitutions per site. (**B**) Representative fluorescence and DIC images showing the subcellular localization (I) and wild-type (WT)-like sporulation (II) of the *S. venezuelae* Δ*sepH* mutant producing SepH$_{Ms}$-mCherry (SS380). For comparison spore chains of the WT and the Δ*sepH* mutant (SV56) are shown and the mean ± SD spore length for each strain is denoted below. 350 spores per biological replicate (n = 3) and strain were measured. Scale bars: 5 μm. (**C**) Yeast two-hybrid analysis to test the interaction between SepH$_{Ms}$ and FtsZ$_{Ms}$ from *M. smegmatis*. Viability of the yeast strains carrying the respective fusion proteins was confirmed by spotting the individual strains on minimal medium lacking leucine and tryptophan (left panel). An interaction between the protein fusions allows growth on minimal medium lacking leucine, tryptophan, histidine, and alanine (right panel). Shown is a representative image. Experiments were performed in triplicate. (**D**) Assembly dynamics of FtsZ$_{Ms}$ from *M. smegmatis* using dynamic light scattering. Light scatter traces for 6 μM FtsZ (black) and 6 μM FtsZ$_{Ms}$ in the presence of 3 μM SepH$_{Ms}$ (red) are shown. 2 mM GTP was added to induce FtsZ polymerization. Light scatter graphs display representative traces of at least three independent experiments. (**E**) FtsZ$_{Ms}$ filament morphology visualized by negative stain TEM of 6 μM FtsZ$_{Ms}$ alone, with 2 mM GTP and with 3 μM SepH$_{Ms}$. SepH$_{Ms}$ (3 μM) does not form visible structures when incubated with GTP. Scale bars: 200 nm. (**F**) High = and low-speed co-sedimentation assay of polymerized FtsZ$_{Ms}$ (6 μM) with and without SepH$_{Ms}$ (3 μM) in the presence of 2 mM GTP. Presence of FtsZ$_{Ms}$ and SepH$_{Ms}$ in the supernatant (S) or pellet (P) was analyzed by SDS-PAGE and Coomassie staining. The average percentage of total FtsZ$_{Ms}$ or SepH$_{Ms}$ in the pellet fraction based on results from two independent experiments is indicated below. Note that due to the number of samples, protein fractions were run on several protein gels which resulted in different staining intensity.

The online version of this article includes the following source data and figure supplement(s) for figure 7:

**Source data 1.** Alignment of SepH proteins used to construct phylogenetic tree and phylogenetic tree file with bootstrap values used to generate *Figure 7A*.
**Source data 2.** Spore measurement data used in *Figure 7B*.
**Source data 3.** Co-sedimentation data used in *Figure 7F*.
**Figure supplement 1.** SepH sequence logo.
**Figure supplement 1—source data 1.** Alignment used to generate SepH sequence logo.
**Figure supplement 2.** SDS gel showing purified SepH$_{Ms}$ and FtsZ$_{Ms}$.
**Figure supplement 3.** Size exclusion chromatogram of purified SepH$_{Ms}$.
**Figure supplement 3—source data 1.** Size exclusion chromatogram data.
**Figure supplement 4.** GTP hydrolysis rate of FtsZ$_{Ms}$ with and without SepH$_{Ms}$.
**Figure supplement 5.** FtsZ$_{Ms}$ (6 μM) filament bundles formed in the presence of 6 μM SepH$_{Ms}$ and 2 mM GTP.

## SepH from *M. smegmatis* stimulates FtsZ polymerization and bundling

To test if SepH$_{Ms}$ can also affect the behavior of FtsZ$_{Ms}$ *in vitro*, we purified recombinant SepH$_{Ms}$-6xHis (SepH$_{Ms}$) and untagged FtsZ$_{Ms}$ (*Figure 7—figure supplement 2*). We first examined SepH$_{Ms}$ by size exclusion chromatography and, like SepH from *S. venezuelae*, it eluted as a single peak that corresponds to the predicted size of a tetramer (148 kDa) (*Figure 7—figure supplement 3*). We also measured the effect of SepH$_{Ms}$ on the GTPase activity of FtsZ$_{Ms}$ but did not observe a significant effect on the GTP turnover rate when SepH$_{Ms}$ was added to the reaction (*Figure 7—figure supplement 4*). Next, we followed the assembly of FtsZ$_{Ms}$ into filaments using DLS. As described previously, mycobacterial FtsZ displayed a low polymerization rate (*White et al., 2000*). However, FtsZ$_{Ms}$ filament assembly was dramatically stimulated upon addition of SepH$_{Ms}$ at a molar ratio of 1:0.5 (*Figure 7D*), as indicated by a sharp and rapid increase in the light scattering signal. Electron microscopy of negatively stained FtsZ$_{Ms}$ (6 μM) confirmed that the incubation with GTP led to the assembly of long and thin protofilaments (*Figure 7E*). Interestingly, and in contrast to SepH from *S. venezuelae*, the addition of 3 μM SepH$_{Ms}$ resulted in the formation of FtsZ$_{Ms}$ bundles, which were even more prominent when FtsZ$_{Ms}$ and SepH$_{Ms}$ were combined at equimolar concentrations (*Figure 7—figure supplement 5*). Notably, in the background of these bundles, shorter FtsZ filaments were visible, suggesting that SepH$_{Ms}$ initially enhanced FtsZ$_{Ms}$ protofilament formation which subsequently led to the assembly of long and stable FtsZ$_{Ms}$ filaments that could further associate into multifilament bundles. This observation was also supported by differential co-sedimentation epxeriments. Incubation of FtsZ$_{Ms}$ with GTP and SepH$_{Ms}$ followed by either high- or low-speed centrifugation resulted in an enrichment of both proteins in the pellet fraction (*Figure 7F*), indicating that SepH$_{Ms}$ led to the formation of macromolecular FtsZ assemblies that can be pelleted at lower

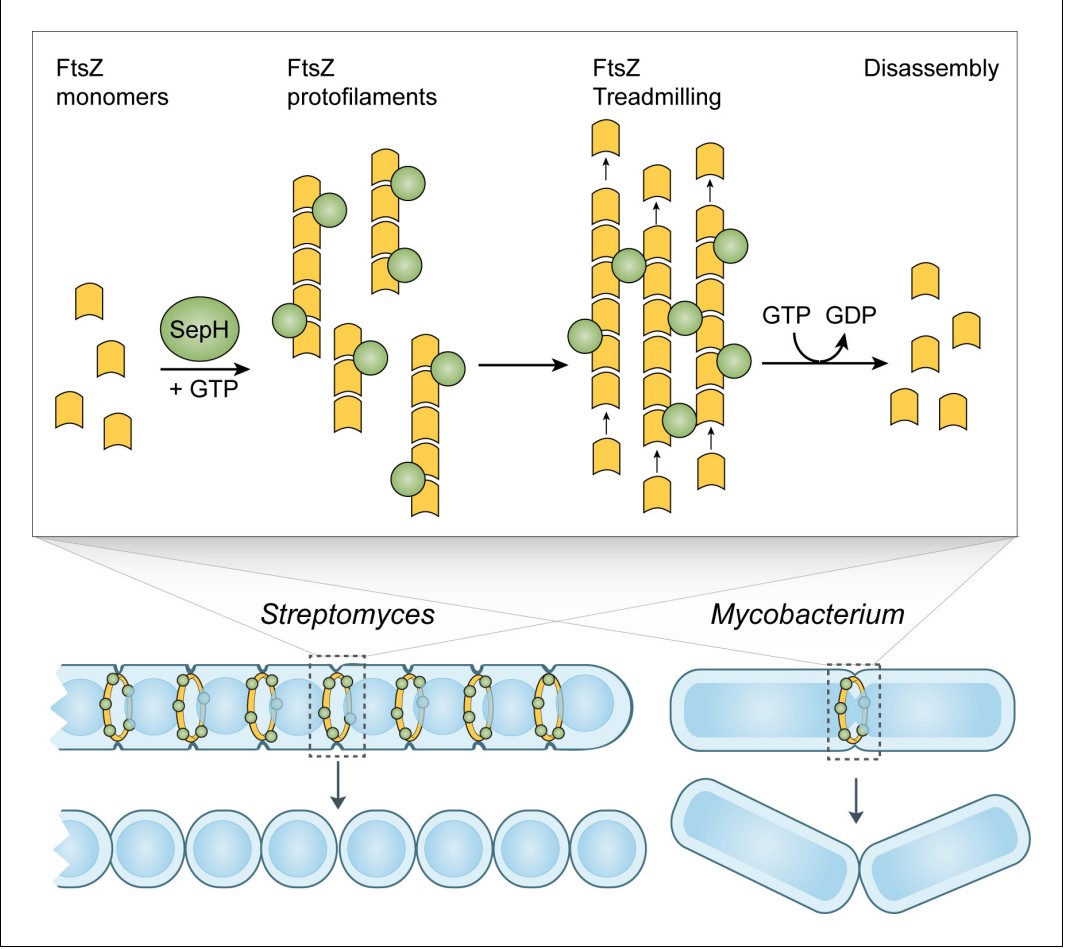

**Figure 8.** Model of SepH-mediated FtsZ remodeling in *Streptomyces* and *Mycobacterium*. SepH (green) directly binds FtsZ (yellow) and stimulates the robust assembly of FtsZ protofilaments. Filament-associated SepH from *M. smegmatis* can further mediate lateral interactions between FtsZ filaments while SepH from *S. venezuelae* is likely to only transiently stabilize FtsZ protofilaments. The GTP hydrolysis rate of FtsZ is likely not directly affected by SepH but will eventually lead to the disassembly of FtsZ filaments. Importantly, SepH functions by increasing the local concentration of FtsZ which promotes the condensation of filaments into a Z-ring and aids FtsZ treadmilling during the early stages of cell division. This process is linked to septal peptidoglycan synthesis and the formation of division septa.

centrifugation rates. This clearly suggests that $SepH_{Ms}$ not only stimulates the rapid polymerization of $FtsZ_{Ms}$ but also has the propensity to promote lateral interactions of $FtsZ_{Ms}$ filaments.

Collectively, our results demonstrate that, similar to SepH from *S. venezuelae,* SepH from *M. smegmatis* not only stimulates but also accelerates the assembly of $FtsZ_{Ms}$ filaments. In addition, $SepH_{Ms}$ further promotes lateral interactions between $FtsZ_{Ms}$ protofilaments, leading to the formation of stable macromolecular FtsZ assemblies *in vitro.*

## Discussion

Here we report the identification of SepH as one of the missing actinomycete-specific positive regulators of Z-ring formation. Based on our *in vivo* and *in vitro* characterization of the SepH homologs from the filamentous species *S. venezuelae* and the rod-shaped species *M. smegmatis,* we propose a model in which SepH-mediated FtsZ assembly increases the local concentration of FtsZ. This in turn promotes the spatial ordering of FtsZ filaments, the formation of division-competent Z-ring(s), and efficient FtsZ treadmilling, which is linked to the synthesis of septal peptidoglycan (*Figure 8*; *Bisson-Filho et al., 2017*; *Yang et al., 2017*).

Our model is supported by several lines of evidence. First, we report that SepH plays a crucial role during the early stages of Z-ring formation. Kymograph analysis of fluorescently tagged FtsZ revealed that individual Z-rings fail to assemble in sporulating *S. venezuelae* hyphae lacking SepH (*Figure 2C*). Notably, this contrasts with earlier results from a *Streptomyces ΔdynAB* mutant in which already assembled Z-rings become destabilized and disassemble, leading to failed septation or partially constricted hyphae (*Schlimpert et al., 2017*). Thus, SepH is clearly important for the establishment of Z-rings. We further found that apart from the irregular spacing, Z-rings assembled in the absence of SepH displayed similar dynamics and architecture to Z-rings in sporulating WT hyphae (*Figure 2E and F*). One possible explanation could be that FtsZ molecules that fail to establish a Z-ring are free to diffuse and to interact with neighboring Z-rings, thereby allowing the WT-like assembly of the remaining Z-rings and compensating for the lack of SepH. In addition, *Streptomyces* undergo a second, distinct mode of division during vegetative growth which leads to the synthesis of cross-walls. Our live-cell imaging studies revealed that SepH is required not only for sporulation-specific cell division but also for vegetative cross-wall formation (*Figure 2G*). The formation of cross-walls is poorly understood and although the synthesis of cross-walls depends on FtsZ, other core cell division proteins such as DivIC, FtsI, FtsL, and FtsW are not absolutely required (*McCormick, 2009*; *Cantlay et al., 2021*). We found that deleting *sepH* significantly reduced the number of cross-walls in vegetative hyphae. The importance of SepH for FtsZ-mediated cell division during vegetative growth was further supported by the observation that *sepH*-deficient hyphae were prone to extensive cell lysis due to reduced hyphal compartmentalization. This is in line with work by Santos-Beneit et al. demonstrating that cross-walls protect the mycelium from large scale cell rupture caused by mechanical or enzymatical stress (*Santos-Beneit et al., 2017*). Thus, despite the different morphological outcomes of the two types of cell division that occur during the *Streptomyces* life cycle, both require SepH for efficient and regular Z-ring formation.

Second, we show that SepH directly interacts with FtsZ *in vitro* and determined the protein domains that are critical for SepH function. Our cytological and biochemical analyses have revealed that the N-terminal DUF3071 domain is crucial for SepH activity during cell division (*Figures 3* and *5*). Importantly, our data support the idea that SepH function depends on a highly conserved HTH motif located within the DUF3071 domain (*Figure 7—figure supplement 1*). Mutational analysis confirmed that this motif is essential for SepH activity in *S. venezuelae* cell division and required for interaction with FtsZ (*Figure 5*). Although the SepH C-terminus is less conserved among SepH homologs and largely dispensable for SepH function, we found that the C-terminal domain (CTD) is required for SepH tetramer formation *in vitro* and efficient subcellular localization *in vivo* (*Figure 5—figure supplement 3B* and *Figure 3*). Furthermore, in the absence of the SepH C-terminal domain, FtsZ filaments assembled together with SepH-NTD appeared to be longer (*Figure 6—figure supplement 2D*), indicating that the CTD influences overall SepH activity. In support of the importance of the C-terminal half for SepH function, we identified two additional short sequence motifs of unknown function within the CTD that are widely conserved among SepH homologs (*Figure 7—figure supplement 1*).

Third, we demonstrate that SepH from *S. venezuelae* and SepH$_{Ms}$ from *M. smegmatis* stimulate FtsZ nucleation and affect FtsZ filament stability which ultimately may aid the formation of division-competent Z-rings *in vivo* (*Figures 6* and *7*). Recent work in *B. subtilis* and *S. aureus* suggest that at the onset of cytokinesis, loose FtsZ filaments are actively condensed into a Z-ring. This process depends on FtsZ treadmilling and the activity of FtsZ-binding proteins, such as SepF, FtsA, ZapA, or GpsB, which support filament formation, bundling, stabilization, or membrane-anchoring (*Eswara et al., 2018*; *Monteiro et al., 2018*; *Silber et al., 2021*; *Squyres et al., 2020*; *Whitley et al., 2020*; *Woldemeskel et al., 2017*). Interestingly, our *in vitro* studies suggest that the SepH homologs from *S. venezuelae* and *M. smegmatis* display biochemical properties that are partially similar to the activities described for ZapA and GspB (*Caldas et al., 2019*; *Eswara et al., 2018*; *Squyres et al., 2020*; *Woldemeskel et al., 2017*). For example, similar to the effect described for GpsB, we found that SepH and SepH$_{Ms}$ promote FtsZ assembly (*Eswara et al., 2018*). In addition, like ZapA, *M. smegmatis* SepH$_{Ms}$ also supports lateral association of FtsZ filaments (*Caldas et al., 2019*) and, under the conditions used, could also induce the formation of FtsZ bundles (*Figure 7e* and *Figure 7—figure supplement 5*). This observation is in line with our comparative GTPase activity assays which suggested that SepH$_{Ms}$ does not affect the GTP turnover rate of FtsZ$_{Ms}$, (*Figure 7—figure supplement 4*). These results are in slight contrast to the observed activity

of SepH from *S. venezuelae*, which led to an increase in the GTPase activity of FtsZ. We believe that the effect on the higher GTP hydrolysis rate of FtsZ is a direct consequence of the more abundant and reversible assembly of FtsZ filaments in the presence of SepH (*Figure 6E and G*). Further structural studies will be required to identify the critical residues at the interface of the SepH-FtsZ complex and to determine whether SepH or SepH$_{Ms}$ association induces a conformational change in the FtsZ structure that could promote the nucleation and/or bundling observed in our *in vitro* experiments. However, the net effect of both SepH homologs leads to a local increase in FtsZ concentration which is likely to also influence FtsZ GTPase activity, thereby mediating efficient FtsZ-treadmilling and Z-ring remodeling. Moreover, as reported for GpsB and ZapA (*Eswara et al., 2018*; *Low et al., 2004*), both characterized SepH homologs form oligomers in solution which could further aid the interaction with FtsZ and the assembly of FtsZ filaments into a condensed Z-ring to drive cytokinesis.

Finally, our combined cytological, biochemical, and phylogenetic analyses support our hypothesis that SepH plays a conserved and crucial role in actinobacterial cell division. Actinobacteria display a range of complex life styles and include human pathogens such as *C. diphtheriae* or *M. tuberculosis*, in which *sepH* is essential (*Barka et al., 2016*; *Griffin et al., 2011*; *Sassetti et al., 2003*). Although SepH is not essential in *S. venezuelae*, *sepH*-deficient hyphae are subject to frequent cell lysis during vegetative growth and sporulate less efficiently (*Video 2*), supporting the notion that SepH plays a critical role in streptomycetes development.

In summary, we propose that SepH functions to promote FtsZ polymerization and in this way orchestrates the assembly, stabilization, and activity of FtsZ at the onset of cell division in actinobacteria.

# Materials and methods

**Key resources table**

| Reagent type (species) or resource | Designation | Source or reference | Identifiers | Additional information |
|---|---|---|---|---|
| Gene (*Streptomyces venezuelae*) | *sepH* | StrepDB | *vnz_27360* | http://strepdb.streptomyces.org.uk/ |
| Gene (*S. venezuelae*) | *ftsZ* | StrepDB | *vnz_08520* | http://strepdb.streptomyces.org.uk/ |
| Gene (*Mycobacterium smegmatis mc² 155*) | *sepH$_{Ms}$* | Mycobrowser | *msmeg_5685* | https://mycobrowser.epfl.ch/ |
| Gene (*M. smegmatis mc² 155*) | *ftsZ$_{Ms}$* | Mycobrowser | *msmeg_4222* | https://mycobrowser.epfl.ch/ |
| Strain, strain background (*S. venezuelae*) | WT | NZ_CP018074.1 | NRRL B-65442 | Wild type |
| Genetic reagent (*S. venezuelae*) | Δ*sepH::apr* | This paper | SV56 | Chromosomal *sepH* locus was replaced by *apr-oriT cassette amplified with* primers mb118/mb119 and then transduced into WT using ΦSV1 |
| Antibody | Anti-SepH (Rabbit polyclonal) | This paper | Cambridge Research Biochemicals | Automated Western blot (1:200) |
| Antibody | Anti-FtsZ (Rabbit polyclonal) | This paper | Cambridge Research Biochemicals | Automated Western blot (1:200) |
| Antibody | Anti-GFP (Rabbit polyclonal) | Sigma Aldrich | SAB4301138-100UL | Automated Western blot (1:200) |
| Recombinant DNA reagent | pTB146 (plasmid) | doi:10.1038/emboj.2008.264 | | Plasmid for heterologous protein production |

*Continued on next page*

*Continued*

| Reagent type (species) or resource | Designation | Source or reference | Identifiers | Additional information |
|---|---|---|---|---|
| Recombinant DNA reagent | pET-21b (plasmid) | Novagen | 69741 | Plasmid for heterologous protein production |
| Recombinant DNA reagent | SepH (pFRL39, plasmid) | This paper | *sepH* in pTB146 | Purification of SepH |
| Recombinant DNA reagent | FtsZ, (pSS287, plasmid) | This paper | *ftsZ* in pTB146 | Purification of FtsZ |
| Recombinant DNA reagent | FtsZ$_{Ms}$ (pSS560, plasmid) | This paper | *ftsZ$_{Ms}$* in pTB146 | Purification of FtsZ$_{Ms}$ |
| Recombinant DNA reagent | SepH$_{Ms}$ (pSS561, plasmid) | This paper | *sepH$_{Ms}$* in pET21b | Purification of SepH$_{Ms}$ |
| Sequence-based reagent | mb118 | This paper | Redirect PCR primer | CACGTGACGTCGGCAGGCA CCACCCGGGAGGTCCCCATGA TTCCGGGGATCCGTCGACC |
| Sequence-based reagent | mb119 | This paper | Redirect PCR primer | AGCCGCGGAACCGGCGGACC GCCACGGCTCCTGCCGTCATGT AGGCTGGAGCTGCTTC |
| Commercial assay or kit | 12–230 KDa Wes separation module | Bio-Techne | SM-W004 | Plate and capillaries for Automated Western blot |
| Commercial assay or kit | WES anti-rabbit detection module | Bio-Techne | DM-001 | Secondary antibody, luminol and reagents for Automated Western blot |
| Commercial assay or kit | Frozen-EZ Yeast Transformation II Kit | Cambridge Bioscience | T2001 | Yeast two-hybrid analysis |
| Commercial assay or kit | Pi ColorLock Kit | Expedeon | 303–0030 | |
| Commercial assay or kit | CellASIC ONIX B04A-03 Microfluidic Bacteria Plate | Millipore | B04A-03-5PK | Time-lapse microscopy |
| Chemical compound, drug | GTP | Jena Bioscience | NU-1012 | GTPase assay, DLS, TEM, Co- sedimentation |
| Chemical compound, drug | GDP | Sigma Aldrich | G7127-10MG | DLS, TEM, Co- sedimentation |
| Chemical compound, drug | GMPCPP (GpCpp) | Jena Bioscience | NU-405S | DLS, TEM, Co- sedimentation |
| Chemical compound, drug | WGA (Wheat Germ Agglutinin), Alexa Fluor 488 Conjugate | Molecular Probes | W11261 | Cell wall staining |
| Chemical compound, drug | 7-AAD (7-Aminoactinomycin D) | Molecular Probes | A1310 | DNA staining |
| Chemical compound, drug | HADA (3-[[(7-Hydroxy-2-oxo-2*H*-1-benzopyran-3-yl) carbonyl]amino] -D-alanine hydrochloride) | Other | | Cell wall staining; Gift from M. Thanbichler: synthesized after doi:10.1038/nprot.2014.197 |
| Software, algorithm | Fiji | Open-source software package | | Image analysis |
| Software, algorithm | ZenBlue 2012 | Zeiss | Version 1.120 | Image analysis |
| Software, algorithm | Compass for SW | Bio-Techne | Version 4.0 | WES |
| Software, algorithm | Prism | GraphPad | Version 9.0 | Data analysis |
| Software, algorithm | CLUSTALX | http://www.clustal.org/clustal2/ | | Phylogenetic analysis |
| Software, algorithm | MAFFT | https://mafft.cbrc.jp/alignment/software/ | | Phylogenetic analysis |

*Continued*

| Reagent type (species) or resource | Designation | Source or reference | Identifiers | Additional information |
|---|---|---|---|---|
| Software, algorithm | MUSCLE | https://www.ebi.ac.uk/Tools/msa/muscle/ | | Phylogenetic analysis |
| Software, algorithm | CD-HIT | http://weizhongli-lab.org/cd-hit/ | | Phylogenetic analysis |
| Software, algorithm | TRIM-AL | http://trimal.cgenomics.org/ | | Phylogenetic analysis |
| Software, algorithm | PHYML | http://www.atgc-montpellier.fr/phyml/ | | Phylogenetic analysis |
| Software, algorithm | iTOL | https://itol.embl.de/ | | Phylogenetic analysis |
| Software, algorithm | WebLogo3 | http://weblogo.threeplusone.com/ | | Sequence logo |

## Bacterial strains and growth conditions

Bacterial strains are listed in *Supplementary file 1* (Table 1). *E. coli* strains were grown in LB or on LB agar at 37℃ supplemented with the following antibiotics when necessary: 100 µg mL$^{-1}$ carbenicillin, 50 µg mL$^{-1}$ kanamycin, 25 µg mL$^{-1}$ hygromycin, 50 µg mL$^{-1}$ apramycin, or 25 µg mL$^{-1}$ chloramphenicol.

*Streptomyces venezuelae* was grown in maltose-yeast extract-malt extract medium (MYM) prepared with 50% tap water and 50% reverse osmosis water and supplemented with R2 trace element solution at 1:500 (*Kieser et al., 2000*). Liquid cultures were grown under aeration at 30℃ at 250 rpm. MYM agar was supplemented with the following antibiotics when required: 5 µg mL$^{-1}$ kanamycin, 25 µg mL$^{-1}$ hygromycin, or 50 µg mL$^{-1}$ apramycin.

Plasmids and oligonucleotides used to generate or to verify strains and plasmids are listed in *Supplementary file 1*, Tables 2 and 3, respectively.

## Construction and complementation of a *sepH* mutant in *S. venezuelae*

Using 'Redirect' PCR targeting (*Gust et al., 2004*; *Gust et al., 2003*), the *sepH* mutant was generated in which the central (1029 bp) coding region was replaced with a single apramycin resistance cassette. A cosmid library that covers >98% of the *S. venezuelae* genome (M.J. Bibb and M.J. Buttner, unpublished) is fully documented at http://strepdb.streptomyces.org.uk/. Cosmid Sv-3-B02 was introduced into *E. coli* BW25113 containing pIJ790 and the *sepH* gene (*vnz_27360*) was replaced with the *apr-oriT* cassette amplified from pIJ773 using the primer pair mb118 and mb119. The resulting disrupted cosmid was confirmed by PCR analysis using the flanking primers mb144 and mb145 and introduced into *S. venezuelae* by conjugation via *E. coli* ET12567/pUZ8002 (*Paget et al., 1999*). Double cross-over strains (Apr$^R$, Kan$^S$) were confirmed by PCR using primer mb144 and mb145. To avoid any unwanted genetic changes following PCR-targeting and homologous recombination, the Δ*sepH::apr* locus was transduced back into WT *S. venezuelae* using the transducing phage SV1 (*Stuttard, 1982*) as described by *Tschowri et al., 2014*. A representative transductant (Apr$^R$) was designated SV56. For complementation, pMB557 was introduced into the *sepH* mutant by conjugation.

## Light microscopy and kymograph analysis

For imaging protein localization in *S. venezuelae*, cells were grown in MYM medium for 14–18 hr and a 2 µL sample of the culture was spotted onto a 1% agarose pad. *Streptomyces* hyphae were visualized using a Zeiss Axio Observer Z.1 inverted epifluorescence microscope fitted with a Zeiss Colibri 7 LED light source and a Zeiss Alpha Plan-Apo 100×/1.46 Oil DIC M27 objective. Still images and time-lapse images series were collected using Zen Blue (Zeiss) and analyzed using Fiji (*Schindelin et al., 2012*).

Time-lapse fluorescence imaging *with S. venezuelae* was performed as previously described (*Schlimpert et al., 2016*). Briefly, *S. venezuelae* strains were grown in MYM medium for about 36 hr at 30℃ and 250 rpm to reach complete sporulation. To isolate spores, mycelium was pelleted at 400× g for 1 min. Supernatants enriched in spores were diluted in MYM medium to a final

concentration of 0.5–5 × 10$^7$ spores/mL. Spores were loaded into B04A microfluidic plates (ONIX, CellASIC), allowed to germinate and grown by perfusing MYM for 3 hr before medium was switched to spent-MYM medium. Spent-MYM was prepared from the 36 hr sporulation culture by filtering the growth medium to remove spores and mycelial fragments. The media flow rate and temperature were maintained at 2 psi and 30℃. Time-lapse imaging was started approximately 8 hr after spores had germinated and images were acquired every 10 min until sporulation was completed.

Kymographs were generated from registered time-lapse image series of strain SS12 (WT/*ftsZ-ypet*) and MB750 (Δ*sepH*/*ftsZ-ypet*) using Fiji (*Schindelin et al., 2012*). Hyphae undergoing sporulation septation were first identified based on the characteristic FtsZ-YPet localization pattern following the cessation of tip extension. 24 frames (10 min/frame) including one frame immediately before and 22 frames after the cessation of hyphal growth were isolated. Selected hyphae were 'straightened' in Fiji and a segmented line (five pt) was manually drawn along the center of the straightened hyphae. FtsZ-YPet fluorescence intensity was plotted along this line as a function of time (240 min) using the 'Reslice' command. Kymographs were further annotated in Adobe Illustrator.

To visualize fluorescence intensities of Z-rings over time, time-lapse series were first corrected for background fluorescence by applying a custom Python script with a multi-Otsu thresholding algorithm. The following steps were performed in Fiji (*Schindelin et al., 2012*): Z-rings were identified manually in time-lapse series and an ROI of 10 × 20 pixels was drawn around each Z-ring. The average fluorescence intensity values within each ROI were then collected and the mean fluorescence intensity trace of all Z-rings isolated from either WT or *sepH*-deficient hyphae was plotted using Graphpad Prism.

To determine the width of Z-rings, an average fluorescence intensity projection for each of the time-series was first generated in Fiji. The corresponding fluorescence intensity trace along a segmented line (five pt) manually drawn along the hyphal midline was then extracted, and the obtained data was further processed in R. For each strain, five independent time-lapse series were analyzed. Peaks (which correspond to potential Z-rings) were identified using a custom R script and further filtered to remove false-positive peaks with a fluorescence intensity below 100. Z-ring width was calculated by measuring the full width at half maximum of the Z-ring peak in the fluorescence intensity profiles. Z-rings widths were further analyzed and plotted using Graphpad Prism.

## Spore length measurements

Lawns of the respective *S. venezuelae* strains were generated by spreading a single colony onto MYM agar. The plates were incubated for 3–4 days at 30℃ until sporulation was completed. Spores were washed off the agar using 20% glycerol and a sterile cotton pad through which spores were harvested using a sterile syringe. A small aliquot of the spore suspension was mounted on a microscope slide on top of a thin agarose pad (1% agarose dissolved in water) and imaged by phase-contrast microscopy using a Zeiss Axio Observer Z.1 inverted microscope and a Zeiss Alpha Plan-Apo 100×/1.46 Oil DIC M27 objective. Spore lengths were determined using the software Fiji (*Schindelin et al., 2012*) except for spore length measurements in *Figure 7B* in which case the MicrobeJ plug-in for Fiji was used (*Ducret et al., 2016*).

## Staining of DNA and peptidoglycan

*S. venezuelae* WT and SV56 cells were grown in confluent patches on MYM agar for 1–2 days. Glass coverslips were gently pressed onto the cell material and removed. Coverslips were fixed with 100% methanol for 1 min. Sterile H$_2$O was used to wash the coverslips. Spore chains attached to the coverslips were incubated with the DNA-stain 7-AAD (7-aminoactinomycin D, 10 µg mL$^{-1}$) and with Wheat Germ Agglutinin (WGA), Alexa Fluor 488 Conjugate (50 µg mL$^{-1}$) to visualize cell wall material. The samples were incubated for 30 min in the dark, after which the dyes were removed with sterile H$_2$O. The coverslips were then mounted onto agarose pads and visualized by fluorescence microscopy.

For HADA (7-hydroxycoumarin 3-carboxylic acid-amino-D-alanine) labeling (*Kuru et al., 2015*), spores were loaded into BA04 microfluidic plates (CellASIC ONIX). Trapped spores were continuously supplied with MYM containing 0.25 mM HADA at 2 psi and 30℃. Following spore germination, hyphae were allowed to grow by perfusing MYM-HADA at 2 psi for 4–5 hr. Prior to image acquisition MYM-HADA was replaced with MYM and hyphae were visualized using fluorescence microscopy

as described above. Images were collected using Zen Blue (Zeiss) and analyzed using Fiji (*Schindelin et al., 2012*).

## Cryo-scanning electron microscopy

*S. venezuelae* colonies were mounted on the surface of an aluminum stub with Tissue Tek OCT (optimal cutting temperature compound) (Agar Scientific Ltd, Essex, UK), plunged into liquid nitrogen slush at approximately −210°C to cryo-preserve the material, and transferred to the cryo-stage of an Alto 2500 cryotransfer system (Gatan, Oxford, England) attached to either a FEI NanoSEM 450 field emission gun scanning electron microscope (FEI Ltd, Eindhoven, The Netherlands) or a Zeiss Supra 55 field emission gun scanning electron microscope (Zeiss UK Ltd, Cambridge). The surface frost was sublimated at −95°C for 3½ min before the sample was sputter coated with platinum for 2 min at 10 mA at below −110°C. Finally, the sample was moved onto the cryo-stage in the main chamber of the microscope, held at approximately −130°C, and viewed at 3 kV.

## Transmission electron microscopy

FtsZ filament morphology was visualized by negative staining and TEM. For FtsZ from *S. venezuelae*, 3.5 µM FtsZ and/or 0.6 µM SepH was prepared in buffer P (50 mM HEPES pH 7.2, 50 mM KCl, 5 mM MgCl$_2$). All the solutions were previously filtered using 0.1 µm centrifugal filter units (Millipore). Reactions were pre-warmed at 30°C for 10 min, started by adding 2 mM GTP, and incubated at 30°C for an additional 15 min. 1 or 3.5 µL of each reaction was placed on a carbon-filmed, 400 mesh copper grid (EM Resolutions, Sheffield, UK) which had been glow discharged for 20 s at 10 mA in an Ace 200 (Leica Microsystems (UK) Ltd, Milton Keynes, UK). After 60 s, excess sample was wicked away using Whatman No. 1 filter paper and grids were negatively stained using 2% (w/v) uranyl acetate in water. Grids were imaged using a Talos F200C transmission electron microscope (Thermo-Fisher Scientific, Eindhoven, The Netherlands) operated at 200 kV, equipped with a 4 k OneView CMOS detector (Gatan UK, Abingdon, Oxfordshire, UK).

For *M. smegmatis* proteins, 6 µM FtsZ$_{Ms}$ was prepared in modified buffer P (50 mM HEPES pH 6.8, 100 mM KCl, 5 mM MgCl$_2$) in the absence or presence of SepH$_{Ms}$ at 3 µM or 6 µM. Reactions were pre-warmed to 37°C for 10 min, and then started by adding 2 mM GTP and incubated for further 20 min. Samples were stained and imaged as described above.

## Automated Western blot analysis

For analysis of protein levels, we used the automated capillary-based immunoassay platform WES (ProteinSimple, San Jose, CA). To prepare proteins samples, 2 mL aliquots of liquid MYM cultures were sampled at the desired time points. Mycelium was pelleted by centrifugation and washed with PBS. Pellets were snap-frozen in liquid nitrogen and stored at −80°C until use. Mycelial pellets were thawed on ice and resuspended in 0.4 mL ice-cold lysis buffer (20 mM Tris pH 8.0, 5 mM EDTA, 1× EDTA-free protease inhibitors [Roche]) and sonicated (5 × 15 s on/15 s off at 5-micron amplitude). Cell lysates were then cleared by centrifugation at 16,000× g for 20 min at 4°C. Total protein concentration was determined using Bradford reagent (Biorad) and 1 µg of total protein was then loaded in technical triplicate into a microplate (ProteinSimple). For the detection of SepH, FtsZ or YPet-fusion proteins anti-SepH antibody (1:200), anti-FtsZ antibody (1:200) or anti-GFP antibody (1:200) was used. Data analysis and the generation of virtual Western blots were done using the Compass Software (Protein Simple, Version XZ).

## Quantification of the cellular molar ratio of FtsZ to SepH

Serial dilutions of purified SepH ($1 \times 10^{-3}$ to $7.8 \times 10^{-6}$ mg mL$^{-1}$) and FtsZ ($5 \times 10^{-4}$ to $3.9 \times 10^{-6}$ mg mL$^{-1}$) were made using 1× polymerization buffer (50 mM HEPES pH 7.2, 50 mM KCl, 5 mM MgCl$_2$) and loaded in technical replicates onto the WES automated blotting system, according to the manufacturer's instructions (ProteinSimple, San Jose, CA). Lysates from sporulating cultures of WT *S. venezuelae* were processed as described above. One microgram of total protein of biological triplicates was loaded into a microplate (ProteinSimple) and FtsZ and SepH were detected with anti-FtsZ anti-SepH antibodies (diluted to 1:200). From the output of the WES, standards for recombinant FtsZ and SepH were quantified and fit with a linear function to create a calibration curve. Using these

calibration curves, the signals generated by FtsZ and SepH from the cell lysates were used to calculate the FtsZ:SepH molar ratio.

## Yeast two-hybrid analysis

The yeast two-hybrid assays were performed in strain *Saccharomyces cerevisiae* Y2HGold (Takara Bio USA). Combination of the two plasmids encoding the desired fusion proteins were transformed into Y2HGold cells using Frozen-EZ Yeast Transformation II Kit (Zymo Research). Selection for growth was carried out on selective drop-out plates lacking leucine and tryptophan (SD$^{-Leu-Trp}$) and single colonies were inoculated into liquid SD$^{-Leu-Trp}$ medium and grown overnight at 30°C. Saturated cultures were diluted 1:4 in water and 5 µL of each dilution was then spotted on SD$^{-Leu-Trp}$ and SD$^{-Leu-Trp-Ade-His}$ (additionally lacking adenine and histidine) in order to test for growth and interaction, respectively. Plates were incubated for 4–5 days at 30°C and scanned. Each interaction was tested in biological triplicate experiments.

## Protein expression and purification

To purify untagged SepH, SepH variants and FtsZ from *S. venezuelae* and FtsZ$_{MS}$ from *M. smegmatis*, *E. coli* Rossetta (DE3) was transformed with derivatives of the plasmid pTB146 to produce His$_6$-SUMO-tagged protein fusions. Cells were grown at 37°C in LB medium containing 50 µg mL$^{-1}$ carbenicillin, 25 µg mL$^{-1}$ chloramphenicol, and 1% glucose overnight and then diluted 1/100 in fresh LB medium containing carbenicillin and chloramphenicol. To induce protein production, 0.5 mM IPTG was added to the culture once cells reached an OD$_{600}$ of 0.5. Cultures were incubated shaking at 30°C for 4 hr and then harvested by centrifugation. Cell pellets were resuspended in Tris-FtsZ buffer (50 mM Tris-HCl pH 8.0, 50 mM KCl, and 10% glycerol) and lysed by sonication for 10 cycles at 15-micron amplitude, 15 s ON, and 30 s OFF. Lysates were centrifuged at 26,000× g for 30 min at 4°C to remove cell debris. His$_6$-SUMO-FtsZ, His$_6$-SUMO-SepH, or His$_6$-SUMO-FtsZ$_{Ms}$ were purified using an HisTrap column in ÄKTA pure (GE Healthcare) and eluted using an increasing concentration of imidazole. Fractions containing protein were pooled and dialyzed overnight at 4°C against Tris-FtsZ buffer containing 1 mM DTT and His$_6$-Upl1 protease at a molar ration of 100:1. The cleaved His$_6$-SUMO tag and His$_6$-Upl1 protease were then removed by incubation with Ni-NTA affinity agarose beads. The flow-through containing untagged FtsZ, SepH, or FtsZ$_{Ms}$ was then concentrated and subjected to size exclusion chromatography on a HiLoad 16/600 Superdex 200 pg column (GE Healthcare) in Tris-FtsZ buffer. Peak protein fractions were pooled and dialyzed overnight against HEPES-FtsZ buffer (50 mM HEPES pH 7.2, 50 mM KCl, and 10% glycerol) and subsequently stored at −80°C until further use.

To purify *M. smegmatis* SepH$_{Ms}$-His$_6$ (SepH$_{Ms}$), *E. coli* Rossetta (DE3) carrying the plasmid pSS561 was induced for protein overexpression and cell lysis was carried out as described above. SepH$_{Ms}$-His$_6$ was purified from lysates using an HisTrap column in ÄKTA pure (GE Healthcare) and eluted using an increasing concentration of imidazole. Fractions containing the protein were pooled and dialyzed overnight against HEPES-FtsZ buffer (50 mM HEPES pH 7.2, 50 mM KCl, and 10% glycerol) and stored at −80°C until use.

## Antibody production

To produce antibodies against FtsZ and SepH from *Streptomyces*, untagged FtsZ and SepH-His$_6$ were overexpressed and purified as described above, and a total amount of 2 mg of purified protein was sent to Cambridge Research Biochemicals (UK) to be used to raise antibodies in rabbits.

## Analytical gel filtration chromatography

Purified SepH, SepH-NTD, SepH-CTD, or SepHG79P was prepared at 30 µM in buffer P (50 mM HEPES pH 7.2, 50 mM KCl, 5 mM MgCl$_2$). A 500 µL sample was subjected to size exclusion chromatography on a Superose 12 10/300 GL column (GE Healthcare) in buffer P using an ÄKTA pure (GE Healthcare) at 0.25 mL min$^{-1}$ constant flow. Gel filtration standards (Bio-Rad) included thyroglobulin (MW 670,000), γ-globuline (MW 158,000), ovalbumin (MW 44,000), myoglobin (MW 17,000), and vitamin B12 (MW 1,350). Standards were separated using the same conditions described above, and the retention volume of each of the proteins was plotted against Log MW. The standard curve was used to calculate the molecular weight of SepH using the retention volume previously obtained. The

same procedure described above was carried out for SepH$_{Ms}$ but using a modified buffer P (50 mM HEPES pH 6.8, 100 mM KCl, 5 mM MgCl$_2$).

## GTPase activity assay

FtsZ GTPase activity was monitored using the PiColorLock Gold kit (Expedeon), a malachite-green-based assay. SepH and FtsZ were diluted to the desired concentration in buffer P (50 mM HEPES pH 7.2, 50 mM KCl, 5 mM MgCl$_2$). The protein solution was incubated for 10 min at 30°C and the reaction was started by adding 50 μM GTP. Samples were taken at 0, 2.5, 5, 7.5, and 10 min. Reactions were stopped by adding an equal volume of 0.6 M perchloric acid. Absorbance at 620 nm was measured and plotted using Microsoft Excel. GTPase activity was determined from the linear range of the curves (*Wasserstrom et al., 2013*). GTPase activity assays for *M. smegmatis* FtsZ$_{Ms}$ and SepH$_{Ms}$ were performed as described above but using a modified buffer P (50 mM HEPES pH 6.8, 100 mM KCl, 5 mM MgCl$_2$) and incubating the protein solutions at 37°C. Samples were taken at 0, 5, 10, 15, and 20 min and data was analyzed as described above.

To determine the critical concentration, the GTPase activity was determined for several FtsZ concentrations (4.5, 3.5, 2.5, and 2 μM) in the absence or presence of 0.6 μM SepH. To accommodate for the number of samples and higher FtsZ concentrations, reactions volumes were reduced to accommodate measurements using 96-well plates and samples were taken every 1.5 min instead of 2.5 min as described above. Each reaction was performed in duplicate. A linear regression was calculated for the GTPase rate with and without SepH to determine the value of the X-intercept (the critical concentration).

## Dynamic light scattering

FtsZ assembly was monitored using a Wyatt Dynapro Titan Dynamic Light Scattering (DLS) instrument. All components of the reaction buffer were filtered using 0.1 μm centrifugal filter units (Millipore). *S. venezuelae* FtsZ (3.5 μM) was prepared in buffer P (50 mM HEPES pH 7.2, 50 mM KCl, 5 mM MgCl$_2$) and SepH was added at the desired concentrations when required. 15 μL of the resulting protein solution was transferred to a quartz cuvette and equilibrated to 30°C for 5 min in the DLS instrument and the laser intensity adjusted until readings reached ~20,000 counts. Baseline readings were taken for 5 min, GTP (50 μM or 2 mM) was added, and light scatter readings were recorded for up to 25 min. The same protocol was followed in the case of GDP or GMPCCP. Data were visualized using Dynamics software (v6), transferred to an Excel file, and plotted using GraphPad Prism. *M. smegmatis* FtsZ$_{Ms}$ (6 μM) was prepared in modified buffer P (50 mM HEPES pH 6.8, 100 mM KCl, 5 mM MgCl$_2$) in the presence or absence of SepH$_{Ms}$ (3 μM). All DLS measurements with *M. smegmatis* proteins were performed at 37°C, baseline readings were first monitored for 5 min, followed by the addition of 2 mM GTP and the recording of the scatter profile for up to 35 min. Data was analyzed as described above.

## CD spectroscopy

SepH or SepHG79P (3.5 μM) were dialyzed overnight against phosphate buffer pH 7.2 to dilute the sodium ions in preparation for CD analysis. Spectra were recorded in 1 nm steps on a Chirascan Plus spectrophotometer (Applied Photophysics) at 20°C in a 0.5 mm quartz cuvette (Hellma). Measurements were collected in triplicate, averaged, and background subtracted with matched buffer using the Chirascan software package. Data were exported to an Excel file and plotted using GraphPad Prism.

## Sedimentation assay

FtsZ (3.5 μM) and/or SepH (0.6 μM) were prepared in buffer P (50 mM HEPES pH 7.2, 50 mM KCl, 5 mM MgCl$_2$). Reactions were incubated at 30°C for 10 min and polymerization was started by adding GTP (2 mM) or GMPCCP (1 mM). Samples were incubated for an additional 15 min at 30°C and then pelleted by ultracentrifugation at 350,000× g for 15 min (high-speed), or at 25,000× g for 30 min (low-speed). Supernatant and pellet fractions were mixed with equivalent volumes of SDS sample buffer. Proteins were visualized by SDS-PAGE and Coomassie staining and protein bands were quantified using Fiji (*Schindelin et al., 2012*).

For *M. smegmatis* proteins, $FtsZ_{Ms}$ (6 µM) and/or $SepH_{Ms}$ (3 µM) were prepared in modified buffer P (50 mM HEPES pH 6.8, 100 mM KCl, 5 mM $MgCl_2$). Reactions were incubated at 37°C for 10 min, started by adding 2 mM GTP final concentration, incubated for an additional 20 min followed by ultracentrifugation and SDS-PAGE analysis as described above.

## Chromatin immunoprecipitation and deep-sequencing

WT *S. venezuelae* and the *ΔsepH* mutant (SV56) were grown in four 30 mL volumes of MYM medium for 18 hr (sporulation). Cross-linking and immunoprecipitation were conducted as described by *Bush et al., 2019* using the anti-SepH polyclonal antibody. Library construction and sequencing were performed by Genewiz (NJ, USA), using Illumina Hiseq (2 × 150 bp configuration, trimmed to 100 bp).

Reads in the fastq files received from the sequencing contractor were aligned to the *S. venezuelae* genome (GenBank accession number CP018074) using the bowtie2 (2) software (version 2.2.9), which resulted in one SAM (.sam) file for each pair of fastq files (paired-end sequencing). For each SAM file, the depth command of samtools (version 1.8) was used to arrive at the depth of sequencing at each nucleotide position of the *S. venezuelae* chromosome (https://www.sanger.ac.uk/science/tools/samtools-bcftools-htslib). From the sequencing depths at each nucleotide position determined in 2, a local enrichment was calculated in a moving window of 30 nucleotides moving in steps of 15 nucleotides as (the mean depth at each nucleotide position in the 30-nt window) divided by (the mean depth at each nucleotide position in a 3000-nucleotide window cantered around the 30-nucleotide window). This results in an enrichment ratio value for every 15 nucleotides along the genome. The enrichment ratios thus calculated were stored in files in the bedgraph format and were used for viewing in IGB. After ensuring good correlation between the replicates (Spearman correlation coefficient >0.95) the mean of the replicates was calculated and used in further calculations. Enrichment in the control was subtracted from the enrichment in the WT files. Significance of enrichment was calculated assuming normal distribution of the control-subtracted enrichment values. The SepH ChIP-seq data has been deposited at the MIAME-compliant ArrayExpress database (https://www.ebi.ac.uk/arrayexpress/) under accession number E-MTAB-9064.

## DNase I footprinting

DNase I footprinting experiments were carried out essentially as previously described (*Bush et al., 2013*) and according to the manufacturer's instructions (Sure Track footprinting kit, Amersham Pharmacia Biotech). DNA fragments from the promoter regions of *vnz35870*, *vnz30075*, and *vnz07520* were amplified by PCR from the PL1_M15, PL1_G3 and PL1_E16 cosmids, using the primer pairs mb1136/mb1129, mb1138/mb1139, and mb1140/mb1133, respectively. Oligonucleotides were first end-labeled with T4 polynucleotide kinase (Amersham Pharmacia Biotech) and [γ-$^{32}$P]-ATP as described by the manufacturer. Binding reactions were carried out at room temperature for 30 min in 1× Polymerization Buffer (50 mM HEPES/KOH pH 7.2, 50 mM KCl, 5 mM $MgCl_2$) in a total volume of 40 µL, and in the presence of approximately 50,000–75,000 cpm of the DNA probe. Following incubation, 10 µL containing 3 units of DNase I (Promega) and 1 µL of $CaCl_2$ was added, mixed, and incubated for 1 min. The reaction was stopped by addition of 140 µL stop solution [192 mM NaAc, 32 mM EDTA, 0.14% SDS, 70 µg yeast-tRNA (Invitrogen)]. Samples were then phenol–chloroform extracted prior to ethanol (96%) precipitation. The pellet was vacuum-dried and resuspended in 5 µL of formamide loading dye (95% formamide, 20 mM EDTA pH 8.0, 0.1% bromophenol blue, 0.1% xylene cyanol FF). 2.5 µL of each sample was loaded on a 6% sequencing gel, next to a G+A ladder, prepared according to the Sure Track footprinting kit (Amersham Pharmacia Biotech). The gel was then vacuum-dried before imaging using image plates, visualized using the FUJIFILM FLA-7000.

## Electrophoretic mobility shift assay

DNA sequences were amplified by PCR using the primer pairs mb1136/mb1129, mb1124/mb1125, and mb1126/mb1127 and the templates PL1_M15, SV-4-G01, and pCOLADuet-1 respectively. This generated probes to test for potential binding of SepH to the promoter region of *vnz35870*, a sequence internal to *vnz08520* (*ftsZ*) and a low-GC sequence from the vector *kanR*-gene (*aphII*). Binding reactions were carried out at room temperature for 30 min in 1× polymerization buffer (50 mM HEPES/KOH pH 7.2, 50 mM KCl, 5 mM $MgCl_2$) in a total volume of 20 µL, and in the

presence of 50 ng of the DNA probe. Following the incubation step, samples were run on pre-cast Mini-PROTEAN TBE gels (Bio-Rad 456–5014) in 0.5× TBE for 60–90 min alongside 100 bp ladder (NEB). Gels were stained for 30 min in ethidium bromide solution before imaging under UV-light.

## Phylogenetics analysis

The SepH sequence from *Streptomyces venezuelae* (*vnz_27360*) was used to BLAST against 3962 representative bacterial species (*Altschul et al., 1997*; *Altschul et al., 1990*; *Camacho et al., 2009*). After reciprocal BLAST analysis and quality filtering, 626 actinobacterial SepH homologs were identified. The 626 sequences were clustered to remove redundancies using CD-HIT at 90% similarity and then clustered again at 75% similarity to reduce the likelihood of misclustering (*Li and Godzik, 2006*). These representative homologs (360 sequences) were used to create three separate sequence alignments using CLUSTALX (*Larkin et al., 2007*), MUSCLE (*Edgar, 2004a*; *Edgar, 2004b*), and MAFFT, using the l-ins-I option (*Katoh and Standley, 2014*). TrimAl was used to compare the alignments for consistency, at which point the most consistent (CLUSTAL) was used and gaps that were present in 80% or more of sequences were trimmed out (*Capella-Gutiérrez et al., 2009*). This alignment was used to generate a tree in PHYML (*Guindon et al., 2010*) using the model, LG +G, as selected by SMS (*Lefort et al., 2017*). The tree was visualized using iTOL, the Interactive Tree of Life (*Letunic and Bork, 2019*). Additionally, the alignment was used to generate a logo using WebLogo3 (*Crooks et al., 2004*).

## Acknowledgements

We would like to thank Kelley Gallagher for helpful discussions on the phylogenetic analysis and Clare Stevenson and Julia Mundy for excellent technical assistance. We thank the JIC Bioimaging facility and staff for technical support and in particular Sergio Lopez for help with image analysis. Work in the lab of JRM was supported by the National Institutes of Health grant GM096268. We gratefully acknowledge funding by the Royal Society (URF\R1\180075) and the BBSRC (BB/T015349/ 1) to SS and for support by the BBSRC Institute Strategic Program grant BBS/E/J000PR9791 to the John Innes Centre.

## Additional information

### Funding

| Funder | Grant reference number | Author |
|---|---|---|
| Royal Society | URF\R1\180075 | Susan Schlimpert |
| Biotechnology and Biological Sciences Research Council | BB/T015349/1 | Susan Schlimpert |
| National Institutes of Health | GM096268 | Joseph R McCormick |
| Biotechnology and Biological Sciences Research Council | BBS/E/J/000PR9791 | Matthew J Bush Susan Schlimpert |

The funders had no role in study design, data collection and interpretation, or the decision to submit the work for publication.

### Author contributions

Félix Ramos-León, Matthew J Bush, Conceptualization, Data curation, Formal analysis, Validation, Investigation, Methodology, Writing - review and editing; Joseph W Sallmen, Data curation, Formal analysis, Validation, Methodology, Writing - review and editing; Govind Chandra, Data curation, Software, Formal analysis; Jake Richardson, Methodology; Kim C Findlay, Methodology, Writing - review and editing; Joseph R McCormick, Conceptualization, Writing - review and editing; Susan Schlimpert, Conceptualization, Resources, Data curation, Formal analysis, Supervision, Funding acquisition, Investigation, Visualization, Methodology, Writing - original draft, Project administration, Writing - review and editing

## Author ORCIDs

Matthew J Bush http://orcid.org/0000-0001-8216-0152
Govind Chandra http://orcid.org/0000-0002-7882-6676
Joseph R McCormick http://orcid.org/0000-0002-9071-7296
Susan Schlimpert https://orcid.org/0000-0001-6364-8056

## Decision letter and Author response

Decision letter https://doi.org/10.7554/eLife.63387.sa1
Author response https://doi.org/10.7554/eLife.63387.sa2

# Additional files

## Supplementary files

• Supplementary file 1. Tables listing bacterial strains, plasmids, and oligonucleotides used in this study.

• Transparent reporting form

## Data availability

ChIP-seq data generated in this study has has been depositited to ArrayExpress database under accession number E-MTAB-9064. All other data genearted or analyzed during this study is included in the manuscript and supporting files. Where considered appropiate, soure data has been uploaded to eLife.

The following dataset was generated:

| Author(s) | Year | Dataset title | Dataset URL | Database and Identifier |
|---|---|---|---|---|
| Ramos-Leon F, Bush MJ, Sallmen JW, Chandra G, Richards J, Findlay KC, McCormick JR, Schlimpert S | 2020 | A conserved cell division protein directly regulates FtsZ dynamics in filamentous and unicellular actinobacteria | https://www.ebi.ac.uk/arrayexpress/experiments/E-MTAB-9064/ | ArrayExpress, E-MTAB-9064 |

The following previously published dataset was used:

| Author(s) | Year | Dataset title | Dataset URL | Database and Identifier |
|---|---|---|---|---|
| Bush MJ, Bibb MJ, Chandra G, Findlay KC, Buttner MJ | 2013 | Genes required for aerial growth, cell division, and chromosome segregation are targets of WhiA before sporulation in *Streptomyces venezuelae* | https://www.ebi.ac.uk/arrayexpress/experiments/E-MTAB-4673/ | ArrayExpress, E-MTAB-4673 |

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
