## [Decision Letter]

**Acceptance summary:**

Bacterial cell division is an intensely studied process as inhibition thereof has potential to form the basis of new antibiotic therapies. However, in actinobacteria, a group of organisms comprising several clinically important bacterial pathogens, this process still requires further investigation. This paper reports the function of SepH in *Streptomycesvenezuelae*, demonstrating that it regulates the assembly of the division machinery responsible for cell constriction and ultimate separation of daughter cells. The study will be of interest to microbiologists and bacterial cell biologists who work on cell cycle and division.

**Decision letter after peer review:**

Thank you for submitting your article "A conserved cell division protein directly regulates FtsZ dynamics in filamentous and unicellular actinobacteria" for consideration by *eLife*. Your article has been reviewed by three peer reviewers, and the evaluation has been overseen by Bavesh Kana as the Senior and Reviewing Editor. The following individual involved in review of your submission has agreed to reveal their identity: Ethan C Garner (Reviewer #2).

The reviewers have discussed the reviews with one another and the Reviewing Editor has drafted this decision to help you prepare a revised submission.

Summary:

Herein, Ramos-Leon and co-workers describe the function of SepH, a highly conserved protein among the actinobacteria that is important for cell division. The model system used is *Streptomycesvenezuelae* (Sven). Like all *Streptomyces*, Sven grows as a filamentous network of cells in a mycelium and undergoes a developmental cycle in which it transitions to growth in aerial hyphae that differentiate into an array of spores. Division is not essential for the vegetative growth of Sven and other *Streptomyces*, but it is essential for spore formation. SepH was originally identified because it is part of the WhiA/WhiB regulon that controls sporulation.

Key findings:

1) Inactivation of SepH results in aberrant spore formation, with the spores being irregularly sized when compared to the WT.

2) SepH was found to localize to the division site in an FtsZ-dependent manner to promote normal Z-ring formation.

3) SepH was also found to interact with FtsZ and to stimulate its polymerization.

4) Importantly, many of these activities were found to be conserved in a mycobacterial homolog of SepH.

Conclusion:

SepH is a key division protein in actinobacteria that functions via its interaction with FtsZ to promote polymerization.

Advance to the field:

As actinobacterial cell division remains poorly characterized relative to the proteobacteria and firmicutes, these results significantly advance the understanding of an essential biological process in this important class of bacteria, which includes major pathogens like *Mycobacterium tuberculosis*.

Essential revisions

Reviewers of your manuscript commented that your paper was clearly articulated and overall, the data presentation style was commendable, resulting in a study of notable scholastic excellence. That said, one part of the paper needs improvement and this relates to the effect of SepH on FtsZ, which is summarized as: SepH accelerates and stimulates FtsZ assembly. Currently, this mechanism is confusing and unclear. The two points of data that evidence an increase in FtsZ polymer are: (1) the observation of SepH increasing GTPase activity, and (2) an increase in light scattering signal. Additional controls in these assays are needed to give some clarity. Both are good measurements but have to be validated with secondary measures to ensure the conclusions are correct, as well as determine the contribution of other complicating factors of each assay. Please address the following

1) With regards to the high amounts of SepH used in the assays relative to FtsZ, for most experiments the authors use equimolar amounts of protein. However, no filament associating protein, save sequesters (profilin, thymosin) exist at or exceed the molar ratio of the monomer. This is important to note, as many filament interacting proteins are seen have different effects at different concentrations – at low levels they may cap, sever, or bundle, but higher levels can artificially stabilize filaments or sequester monomers. Thus, the authors should determine the cellular levels of SepH by Western blot (for which they seem to have antibodies, and maybe already this data).

2) Once SepH levels are known, the authors should redo some of the *in vitro* experiments at the cellular concentration of SepH. At the least, redo the key biochemical experiments (detailed below) at a lower concentration of SepH – such as 0.6 um, which appears to be already an amount needed for the maximal increase in their GTPase assay.

3) If and how SepH accelerates and stimulates assembly should be clarified, as how this would occur is currently contradictory and unclear. There are a few different ways SepH could increase FtsZ dynamics, but clarity between increasing the initial assembly rate (nucleation) and turnover (polymer dynamics at equilibrium) have to be separated for the proposed mechanism to be clear. It should be noted that it appears highly unlikely that SepH accelerates FtsZ filament nucleation: if this were the case, SepH would have to increase FtsZ nucleation rate. However, FtsZ is known to rapidly nucleate, making this an unfavourable step for kinetic regulation, and moreover, the data in Figure 6—figure supplement 2 panel C speaks against this hypothesis. This figure shows that FtsZ rapidly nucleates and that SepH has no effect on this first nucleating phase of polymerization. Therefore, the phrases "promotes assembly" and "accelerates" should be modified or removed, as these imply an increase in the rate of polymer formation from pure monomer, which at this stage, is not supported by the data. If the authors wish to test the "accelerates" claim, they should measure the initial phase of FtsZ polymerization by light scattering in a rapid mixer (preferably near FtsZ's critical concentration, for the most dynamic range) whilst slowly increasing the amount of SepH in solution, starting from low nanomolar upward.

4) It must also be noted that measurements of the initial phase of polymerization should be assayed with hydrolyzable GTP, as nucleotide analogs are known to cause kinetic oddities in a wide number of polymers. Due to these effects, for the longer traces – it is preferred if the authors showed the effects of saturating GTP in the main text

5) It is obvious that SepH increases FtsZ GTPase activity, but this assay alone makes it hard to conclude any mechanism, as GTPase activity is not a direct readout of polymer dynamics. This is because FtsZ only hydrolyzes GTP in the polymer form. Thus, GTPase rates are a convolution of the amount of polymer and the associated polymer dynamics. However, from the current data, it is not clear how much of this increase comes from SepH causing an increase in FtsZ polymer dynamics or a change in the total amount of FtsZ polymer (critical concentration). Therefore, to strengthen the conclusions, the authors should determine if SepH changes the critical concentration of FtsZ. This could be done by incrementally diluting the total amount of FtsZ within the reaction, as they monitor the GTPase activity. Plotting the GTPase rate (or the plateau value of the light scattering signals) vs total FtsZ concentration will make a line where the X intercept is the critical concentration. Doing this FtsZ titration in the presence of a small amount of SepH (enough to see an effect in the GTPase assay), compared to the same titration without SepH, will give clarity into the underlying cause of the difference in GTPase rates. If there is no change to the critical concentration, SepH must be somehow changing turnover. If SepH lowers the critical concentration, then a mechanism for "promoting assembly" can be concluded.

6) The experiments with low speed pelleting and EM are not adequate to claim that SepH does not bundle filaments in one species, while it does indeed bundle FtsZ using proteins from another species. This concern arises from the relative timescales of when the low speed pelleting and EM bundling assays were conducted relative to the kinetics observed in the light scattering curves. Critically, light scattering reads filament formation and filament bundling. Work with both FtsZ and other polymers has shown that while filament formation may be fast, most often filament bundling (and the resultant large increase in light scattering) is a slower process, occurring after the initial burst. Thus, the question of SepH promoting bundling of FtsZ becomes highly dependent on the time when the bundling is assayed by EM or slow speed pelleting. The relative difference in the timing of these 2 processes, depending on when they were assayed, could lead to this organismal discrepancy. For the EM assays, the Materials and methods indicate samples were combined and incubated for 10 minutes before spotting onto a grid. For pelleting, it states samples were incubated for 15 minutes. When both of these timescales are placed into the context of the polymerization curves, a possible discrepancy arises:

For *M. smegmatis* SepH and FtsZ: looking at the polymerization curve in Figure 7D, it is clear there is a huge jump (and divergence between curves) in the light scattering in the FtsZ + SepH curves around ~6 minutes, a time before the EM and pelleting timepoints were taken (10 and 15 minutes). As the EM and the pelleting indicate filaments are bundled at this time point, the difference in the curves would suggest the increase in light scattering is, in some part, due to an increase in filament bundling. However, if the same comparison is drawn looking at the *S. venezuelae* polymerization curve (Figure 6—figure supplement 2 panel C), the conclusion that SepH does not bundle FtsZ becomes questionable. The light scattering curve shows that FtsZ alone and FtsZ + SepH both show an initial burst and small overshoot in the 2 minutes, most likely the initial burst of nucleation and equilibration of the polymer. However, the longer phase of the light scattering increase (most likely bundling) takes far longer compared to M. smegmatis, only beginning to take off around 20 minutes, and not beginning to plateau until 40 minutes. Critically, the assays of bundling by pelleting and EM were conducted at 10 and 15 minutes into the reaction, far before the light scattering curve indicates filament bundling might be occurring.

Thus, in order to test (or verify) the conclusion that *S. venezuelae* SepH does not bundle FtsZ, the authors should repeat the EM and slow speed pelleting assays at a later timepoint, one matching a timescale of a large signal increase in the light scattering data (30 or 40 minutes), using identical concentrations of proteins and stating hydrolyzable GTP in all assays.

7) The observation that SepH inhibits ring formation is well noted. However, is there an effect on the dynamics (ring construction) in the rest of the septa? This should be easily quantifiable from the existing data.

8) Figure 2 and related text, the authors show some interesting kymographs +/- SepH and relate them to dynamics. It is not clear what these mean with regards to dynamics. Clearly the spacing is affected by the absence of SepH; however, the temporal related claims are not very convincing. It would be preferable if they could quantify these claims of gaps-ladders.

9) Figure 6—figure supplement 2 panel A – There appears to be some problem (perhaps air bubbles) in the GDP polymerization curve of FtsZ alone in Figure 6—figure supplement 2 panel A. This experiment should be redone.

---

## [Author Response]

Essential revisionsReviewers of your manuscript commented that your paper was clearly articulated and overall, the data presentation style was commendable, resulting in a study of notable scholastic excellence. That said, one part of the paper needs improvement and this relates to the effect of SepH on FtsZ, which is summarized as: SepH accelerates and stimulates FtsZ assembly. Currently, this mechanism is confusing and unclear. The two points of data that evidence an increase in FtsZ polymer are: (1) the observation of SepH increasing GTPase activity, and (2) an increase in light scattering signal. Additional controls in these assays are needed to give some clarity. Both are good measurements but have to be validated with secondary measures to ensure the conclusions are correct, as well as determine the contribution of other complicating factors of each assay. Please address the following1) With regards to the high amounts of SepH used in the assays relative to FtsZ, for most experiments the authors use equimolar amounts of protein. However, no filament associating protein, save sequesters (profilin, thymosin) exist at or exceed the molar ratio of the monomer. This is important to note, as many filament interacting proteins are seen have different effects at different concentrations – at low levels they may cap, sever, or bundle, but higher levels can artificially stabilize filaments or sequester monomers. Thus, the authors should determine the cellular levels of SepH by Western blot (for which they seem to have antibodies, and maybe already this data).

We thank the reviewers for raising this critical point and fully agree with the concerns raised. As suggested, we performed quantitative Western blotting to determine the cellular levels of FtsZ and SepH. Due to the multicellular growth mode of *Streptomyces* filaments, which consist of compartments of variable lengths (e.g. see fluorescence micrographs in Figure 2E for the WT), it is difficult to definitively calculate the cellular concentration of SepH and FtsZ. However, this analysis revealed that the intracellular molar ratio of SepH to FtsZ is 1:6. We have added this new information in the revised manuscript (L288-L293 and Figure 5—figure supplement 4).

Notably, we reported in our original manuscript that using a 1:6 ratio of SepH (0.6 μM) to FtsZ (3.5 μM) was sufficient to stimulate GTPase activity and filament assembly of FtsZ, suggesting that the observed effects on FtsZ behavior are physiologically relevant. We have now repeated multiple key *in vitro* experiments using SepH and FtsZ at a molar ratio of 1:6 (see below).

2) Once SepH levels are known, the authors should redo some of the in vitro experiments at the cellular concentration of SepH. At the least, redo the key biochemical experiments (detailed below) at a lower concentration of SepH – such as 0.6 um, which appears to be already an amount needed for the maximal increase in their GTPase assay.

In light of the results obtained to address comment 1, we have repeated the following biochemical experiments: (1) dynamic light scattering, (2) co-sedimentation and (3) TEM imaging using protein concentrations that reflect the determined molar ratio of both proteins *in vivo*. Specifically, we used 3.5 μM FtsZ and 0.6 μM SepH in our revised *in vitro* reactions. In summary, and in agreement with our original results, we were able to confirm that SepH from *S. venezuelae* has a positive effect on FtsZ polymerization.

1) Dynamic light scattering experiments

We confirmed that SepH affects FtsZ polymerization and led to an increase in light scattering in the presence of 2 mM GTP even when present at a 6-fold lower molar ratio. Furthermore, using the revised reaction conditions, we found that the FtsZ polymerization reactions already reached a plateau after 10 min following the incubation with GTP and SepH. Therefore, we decided to use a 15-minute incubation period of FtsZ with GTP and SepH prior to co-sedimentation and protein negative stain TEM analyses.

In addition, using the 1:6 molar ratio, we recorded light scattering traces of FtsZ polymerization using different nucleotides (2 mM GDP and 2 mM GMPCCP) and in the presence of the different SepH variants (SepH-NTD, SepH-CTD, SepH-G79P). In summary, we only observed a moderate increase in light scattering when the FtsZ polymerization reactions contained SepH-NTD, which is consistent with our earlier results. In contrast to our previous DLS results using SepH and FtsZ at equimolar amounts, incubating FtsZ with a 6-fold lower concentration of SepH and the slow-hydrolysable GTP analogue GMPCCP did not markedly affect the assembly kinetics of FtsZ. To confirm that SepH can interact with FtsZ filaments in the absence of GTP hydrolysis, we performed additional co-sedimentation experiments (point (2)).

In line with the new data, we have modified the manuscript accordingly and replaced the respective panel in Figure 6 with new DLS graphs using the revised reaction conditions.

2) Co-sedimentation experiments

We repeated all high-speed co-sedimentation experiments presented in the original manuscript using 3.5 μM FtsZ (with or without 2 mM GTP) and in the presence or absence of 0.6 μM SepH or SepH-G79P from *S. venezuelae*. Based on our new DLS results (see above), each reaction was incubated for 15 min at 30°C prior to ultracentrifugation. In agreement with our original sedimentation results, we found that the addition of SepH led to a consistent increase of polymerized FtsZ in the pellet fraction from 35% (no SepH) to 47% (with SepH). Incubation of FtsZ with GTP and the SepH helix-turn-helix mutant variant SepH-G79P resulted in FtsZ levels in the pellet fraction (36%) similar to reactions in which SepH was omitted (35 %). The amount of pelleted FtsZ doubled when polymerization was induced with the slow-hydrolysable GTP analogue GMPCCP (84%), which was also independent of SepH. Additionally, while wild-type SepH almost completely co-sedimented with FtsZ in the presence of both GTP (97%) and GMPCCP (98%), it was largely soluble in the absence of FtsZ. Taken together, these new sedimentation patterns are consistent with our previous results, showing that SepH directly interacts with FtsZ (at 1:6 molar ratio) and that this interaction does not require FtsZ to hydrolyze GTP.

We have updated the manuscript and the corresponding figure panels (e.g. Figure 5F and Figure 5—figure supplement 5).

3) Protein negative stain TEM

We have repeated the visualization of FtsZ filaments assembled using GTP or GMPCCP and in the presence of either wild-type SepH or the SepH variants (SepH-NTD, SepH-CTD, SepH-G79P). Each reaction mix was incubated for 15 min at 30C prior to negative staining and imaging. Using these conditions, we confirmed that FtsZ filaments appeared to be more abundant but similar in morphology when SepH was present. These findings are consistent with the results obtained in the DLS experiments and co-sedimentation assays, showing that SepH stimulates the assembly of FtsZ filaments but does not appear to actively promote FtsZ bundling.

We have modified the main text where necessary and replaced the original TEM images in Figure 6A-D and Figure 6—figure supplement 2 with images collected using less SepH (0.6 uM) and consistent reaction conditions. We have removed Figure 6E showing the width distribution of FtsZ filaments assembled with and without SepH because this analysis was based on data obtained using the original (higher) molar ratio of FtsZ and SepH at 1:1.

3) If and how SepH accelerates and stimulates assembly should be clarified, as how this would occur is currently contradictory and unclear. There are a few different ways SepH could increase FtsZ dynamics, but clarity between increasing the initial assembly rate (nucleation) and turnover (polymer dynamics at equilibrium) have to be separated for the proposed mechanism to be clear. It should be noted that it appears highly unlikely that SepH accelerates FtsZ filament nucleation: if this were the case, SepH would have to increase FtsZ nucleation rate. However, FtsZ is known to rapidly nucleate, making this an unfavourable step for kinetic regulation, and moreover, the data in Figure 6—figure supplement 2 panel C speaks against this hypothesis. This figure shows that FtsZ rapidly nucleates and that SepH has no effect on this first nucleating phase of polymerization. Therefore, the phrases "promotes assembly" and "accelerates" should be modified or removed, as these imply an increase in the rate of polymer formation from pure monomer, which at this stage, is not supported by the data. If the authors wish to test the "accelerates" claim, they should measure the initial phase of FtsZ polymerization by light scattering in a rapid mixer (preferably near FtsZ's critical concentration, for the most dynamic range) whilst slowly increasing the amount of SepH in solution, starting from low nanomolar upward.

We agree with the reviewer’s that the data in the original manuscript do not provide sufficient evidence that SepH affects the nucleation step of FtsZ filament formation However, in response to reviewer comment 5 (see below), we found that in the presence of 0.6 μM SepH, the critical concentration of FtsZ is clearly reduced from 1.43 to 1.11 μM. This supports our claim that SepH positively influences FtsZ filament formation and thus, we believe that stating “SepH promotes the assembly of FtsZ filaments” is appropriate. Since we did not have the capacity to probe the effect of SepH on FtsZ polymer dynamics, we have removed any phrases implying that SepH from *S. venezuelae* affects the speed of FtsZ filament formation.

4) It must also be noted that measurements of the initial phase of polymerization should be assayed with hydrolyzable GTP, as nucleotide analogs are known to cause kinetic oddities in a wide number of polymers. Due to these effects, for the longer traces – it is preferred if the authors showed the effects of saturating GTP in the main text

Agreed. We now show the results of our revised DLS experiments using excess amounts of GTP (2 mM) in the main text in Figure 6F and DLS traces obtained from experiments with the GTP analogue GMPCCP in the figure supplement (Figure 6 —figure supplement 3).

5) It is obvious that SepH increases FtsZ GTPase activity, but this assay alone makes it hard to conclude any mechanism, as GTPase activity is not a direct readout of polymer dynamics. This is because FtsZ only hydrolyzes GTP in the polymer form. Thus, GTPase rates are a convolution of the amount of polymer and the associated polymer dynamics. However, from the current data, it is not clear how much of this increase comes from SepH causing an increase in FtsZ polymer dynamics or a change in the total amount of FtsZ polymer (critical concentration). Therefore, to strengthen the conclusions, the authors should determine if SepH changes the critical concentration of FtsZ. This could be done by incrementally diluting the total amount of FtsZ within the reaction, as they monitor the GTPase activity. Plotting the GTPase rate (or the plateau value of the light scattering signals) vs total FtsZ concentration will make a line where the X intercept is the critical concentration. Doing this FtsZ titration in the presence of a small amount of SepH (enough to see an effect in the GTPase assay), compared to the same titration without SepH, will give clarity into the underlying cause of the difference in GTPase rates. If there is no change to the critical concentration, SepH must be somehow changing turnover. If SepH lowers the critical concentration, then a mechanism for "promoting assembly" can be concluded.

This is a good point and we thank the reviewers for their detailed suggestions. To strengthen our conclusion that SepH positively influences FtsZ polymer formation, we have determined the critical concentration for GTPase activity of FtsZ alone and in the presence of 0.6 μM SepH. We previously showed that this SepH concentration is sufficient to increase the GTP hydrolysis rate of FtsZ (see also detailed response to comment 2). In our hands, FtsZ displayed a critical concentration of 1.43 μM. Importantly, addition of SepH reduced the critical concentration of FtsZ to 1.11 μM, supporting the idea that SepH from *S. venezuelae* promotes FtsZ filament assembly *in vitro*.

We have added the new data to the revised manuscript (L369-375 and Figure 6G).

6) The experiments with low speed pelleting and EM are not adequate to claim that SepH does not bundle filaments in one species, while it does indeed bundle FtsZ using proteins from another species. This concern arises from the relative timescales of when the low speed pelleting and EM bundling assays were conducted relative to the kinetics observed in the light scattering curves. Critically, light scattering reads filament formation and filament bundling. Work with both FtsZ and other polymers has shown that while filament formation may be fast, most often filament bundling (and the resultant large increase in light scattering) is a slower process, occurring after the initial burst. Thus, the question of SepH promoting bundling of FtsZ becomes highly dependent on the time when the bundling is assayed by EM or slow speed pelleting. The relative difference in the timing of these 2 processes, depending on when they were assayed, could lead to this organismal discrepancy. For the EM assays, the Materials and methods indicate samples were combined and incubated for 10 minutes before spotting onto a grid. For pelleting, it states samples were incubated for 15 minutes. When both of these timescales are placed into the context of the polymerization curves, a possible discrepancy arises:For M. smegmatis SepH and FtsZ: looking at the polymerization curve in Figure 7D, it is clear there is a huge jump (and divergence between curves) in the light scattering in the FtsZ + SepH curves around ~6 minutes, a time before the EM and pelleting timepoints were taken (10 and 15 minutes). As the EM and the pelleting indicate filaments are bundled at this time point, the difference in the curves would suggest the increase in light scattering is, in some part, due to an increase in filament bundling. However, if the same comparison is drawn looking at the S. venezuelae polymerization curve (Figure 6—figure supplement 2 panel C), the conclusion that SepH does not bundle FtsZ becomes questionable. The light scattering curve shows that FtsZ alone and FtsZ + SepH both show an initial burst and small overshoot in the 2 minutes, most likely the initial burst of nucleation and equilibration of the polymer. However, the longer phase of the light scattering increase (most likely bundling) takes far longer compared to M. smegmatis, only beginning to take off around 20 minutes, and not beginning to plateau until 40 minutes. Critically, the assays of bundling by pelleting and EM were conducted at 10 and 15 minutes into the reaction, far before the light scattering curve indicates filament bundling might be occurring.Thus, in order to test (or verify) the conclusion that S. venezuelae SepH does not bundle FtsZ, the authors should repeat the EM and slow speed pelleting assays at a later timepoint, one matching a timescale of a large signal increase in the light scattering data (30 or 40 minutes), using identical concentrations of proteins and stating hydrolyzable GTP in all assays.

We thank the reviewers for pointing out the possible discrepancy associated with our experimental design. To resolve this, we have performed additional experiments to clarify if SepH from *S. venezuelae* can bundle FtsZ protofilaments. We now present results obtained using identical protein concentrations and consistent timescales for TEM experiments and sedimentation assays. Specifically, we used 3.5 μM FtsZ, 2 mM GTP and 0.6 μM SepH. Because of using a lower, physiologically more relevant, SepH concentration, our revised DLS studies revealed that FtsZ polymerization reached a steady-state level much quicker (<10 min) compared to the conditions used in our original DLS experiments (see Figure 6E revised manuscript). Consequently, we have repeated the low-speed co-sedimentation studies and TEM analyses of FtsZ filament morphology with and without SepH following a 15 min incubation time.

1) Low speed co-sedimentation

With or without SepH, we consistently found the majority of polymerized in the supernatant fraction, indicating the absence of large molecular FtsZ assemblies.

2) TEM of FtsZ filaments

In line with the low-speed sedimentation assays, inspection of the electron micrographs showed that in the presence of SepH, FtsZ assembled into curved protofilaments that were randomly distributed across the EM grid. We did not observe FtsZ bundles following an incubation time of 15 min. As pointed out by the reviewers earlier, we note that the effect of SepH on FtsZ filament morphology is concentration-dependent and that when using a higher SepH concentration (e.g. an equimolar ratio of FtsZ and SepH), FtsZ filaments became much better discernible on EM grids and were occasionally organized into thin bundles (Figure 6—figure supplement 2). We also took images following a 30 min incubation period but did not observed a significant increase of these aggregates (Author response image 1). Given the frequency of these FtsZ filament assemblies and the fact that the cellular concentration of SepH is likely closer to being 6-fold lower than FtsZ (see also response to comment 1), we believe that any possible bundling activity of *S. venezuelae* SepH should be considered with caution and the thin bundles observed in these images are likely a by-product of the artificial stabilization of FtsZ filaments by SepH.

**Author response image 1. sa2fig1:** Representative TEM images of negative stained FtsZ polymers after 30 min of incubation with 2 mM GTP and in the absence or presence of either 0. 6 μM SepH (6:1) or 3.5 μM SepH (1:1). Scale bar: 200 nm.

We have updated the respective panels in Figure 6 and the corresponding figure supplements and discussed the new results in the revised manuscripts.

7) The observation that SepH inhibits ring formation is well noted. However, is there an effect on the dynamics (ring construction) in the rest of the septa? This should be easily quantifiable from the existing data.

We would like to emphasize that SepH does not inhibit Z-ring formation. Our results support the idea that SepH is required for the efficient and regular assembly of division-competent Z-rings during sporulation septation. However, the reviewer raised an interesting point that we had not investigated.

To determine if the absence of SepH affects the formation of the remaining Z-rings in sporogenic hyphae, we first extracted the fluorescence intensity of Z-rings detected in wild-type and ∆*sepH* mutant hyphae from the same time-lapse imaging series that were used to generate the kymographs shown in Figure 3 and the corresponding figure supplements. These intensity traces were averaged to create a mean intensity trace for Z-rings assembled in wild-type and ∆*sepH* hyphae. Analysis of the resulting graph indicates that Z-rings seem to appear slightly earlier in *sepH*-deficient hyphae. However, the overall dynamics of Z-ring assembly, constriction and disassembly, which are reflected by a steady increase of fluorescence intensity and subsequent decrease in fluorescence as Z-rings disassemble, is highly similar between both strains.

In addition, we determined the average width of Z-rings assembled in wild-type and ∆*sepH* mutant hyphae. For this we first generated an average intensity profile of FtsZ-YPet fluorescence for each of the available time-lapse imaging series. Intensity peaks in these profiles, which correspond to Z-rings, were detected automatically and the full width of identified Z-rings was calculated at the half maximum intensity of each peak. Subsequently, we plotted the total distribution of peak widths for each strain and replicate (n=5). This analysis revealed that there is no marked difference between widths of Z-rings assembled in ∆*sepH* mutant hyphae compared to wildtype. It is conceivable that in the absence of SepH, FtsZ molecules that fail to polymerize into a Z-ring are free to interact with neighboring Z-rings, thereby supporting the wildtype-like assembly of the remaining Z-rings.

Taken together, under the employed conditions we did not detect any severe defects in the assembly dynamics and the overall architecture of Z-rings in the absence of SepH.

We have included this new data in the revised manuscript (L158-162), added two new panels to Figure 3 and provided details about the image analysis in the Materials and methods section.

8) Figure 2 and related text, the authors show some interesting kymographs +/- SepH and relate them to dynamics. It is not clear what these mean with regards to dynamics. Clearly the spacing is affected by the absence of SepH; however, the temporal related claims are not very convincing. It would be preferable if they could quantify these claims of gaps-ladders.

To strengthen the conclusions drawn from these kymographs, we have extracted and plotted the average fluorescence intensity of Z-rings over time (please see also detailed response to comment 7 above). The resulting intensity graphs, now presented in the revised manuscript in Figure 3E, clearly show an increase and decrease of FtsZ-YPet fluorescence intensity over time. Comparing the average intensity traces of Z-rings detected in wild-type and *sepH*-deficient hyphae revealed that the overall temporal pattern of FtsZ-YPet fluorescence is similar in both strains and does not indicate any major defects in the dynamics of Z-ring formation associated with the absence of SepH in the remaining Z-rings.

In addition, we quantified the importance of SepH for the regular spacing of Z-rings. In line with our spore-size measurements (Figure 1—figure supplement 2), this analysis confirmed that the absence of SepH led to a higher variability in the spacing of Z-rings (Author response image 2), with Z-rings being assembled every 1.21±0.1 μm in the wildtype (spore size: 1.2±0.23 μm) compared to 1.51±0.2 μm in the ∆*sepH* mutant strain (spore size: 1.4±0.45 μm). We agree with the reviewer that the different spacing between Z-rings in *sepH*-deficient hyphae is obvious in the kymographs. Since the analysis of the distance between Z-rings in sporogenic hyphae did not provide any further information, we opted not to add a separate panel to the Figure 2 supplement.

**Author response image 2. sa2fig2:** Average distance between Z-rings in wild-type and ∆sepH mutant hyphae. Results are based on data extracted from 5 independent time-lapse series of sporulating wild-type and sepH-deficient hyphae, respectively, which were also used to generate kymographs shown in Figure 2 and the corresponding figure supplements (main manuscript). Grey dots present entire data set, colored dots are the mean distance for each experimental replicate, line and bars show total mean distance with 95% CI.

9) Figure 6—figure supplement 2 panel A – There appears to be some problem (perhaps air bubbles) in the GDP polymerization curve of FtsZ alone in Figure 6—figure supplement 2 panel A. This experiment should be redone.

Agreed. We have repeated this experiment and replaced the original DLS traces with new light scattering traces, confirming the clear absence of FtsZ polymerization in the presence of 2 mM GDP (Figure 6—figure supplement 3A).